# An increase in reactive oxygen species underlies neonatal cerebellum repair

Anna Pakula[1†], Salsabiel El Nagar[1†], N Sumru Bayin[1,2,3†], Jens Bager Christensen[2,3], Daniel Stephen[1], Adam James Reid[2], Richard P Koche[4], Alexandra L Joyner[1,2,5]*

[1]Developmental Biology Program, Sloan Kettering Institute, New York, United States; [2]Gurdon Institute, Cambridge University, Cambridge, United Kingdom; [3]Department of Physiology, Development and Neuroscience, Cambridge University, Cambridge, United Kingdom; [4]Center for Epigenetics Research, Memorial Sloan Kettering Cancer Center, New York, United States; [5]Biochemistry, Cell and Molecular Biology Program and Neuroscience Program, Weill Cornell Graduate School of Medical Sciences, New York, United States

*For correspondence:
joynera@mskcc.org

†These authors contributed equally to this work

## eLife Assessment

This **important** work substantially advances our understanding of reactive oxygen species (ROS) as a regenerative signal during postnatal cerebellum repair by activating adaptive progenitor reprogramming. The evidence supporting the conclusions is **compelling**, with rigorous genomic assays and in vivo analyses. This work will be of broad interest to biologists working on stem cells, neurodevelopment and regenerative medicine.

**Abstract** The neonatal mouse cerebellum shows remarkable regenerative potential upon injury at birth, wherein a subset of Nestin-expressing progenitors (NEPs) undergoes adaptive reprogramming to replenish granule cell progenitors that die. Here, we investigate how the microenvironment of the injured cerebellum changes upon injury and contributes to the regenerative potential of normally gliogenic-NEPs and their adaptive reprogramming. Single-cell transcriptomic and bulk chromatin accessibility analyses of the NEPs from injured neonatal cerebella compared to controls show a temporary increase in cellular processes involved in responding to reactive oxygen species (ROS), a known damage-associated molecular pattern. Analysis of ROS levels in cerebellar tissue confirms a transient increase 1 day after injury at postnatal day 1, overlapping with the peak cell death in the cerebellum. In a transgenic mouse line that ubiquitously overexpresses human mitochondrial catalase (mCAT), ROS is reduced 1 day after injury to the granule cell progenitors, and we demonstrate that several steps in the regenerative process of NEPs are curtailed, leading to reduced cerebellar growth. We also provide preliminary evidence that microglia are involved in one step of adaptive reprogramming by regulating NEP replenishment of the granule cell precursors. Collectively, our results highlight that changes in the tissue microenvironment regulate multiple steps in adaptive reprogramming of NEPs upon death of cerebellar granule cell progenitors at birth, highlighting the instructive roles of microenvironmental signals during regeneration of the neonatal brain.

## Introduction

The microenvironment surrounding a brain injury and the cellular responses elicited in the remaining cells are key determinants of how efficiently a repair process will unfold. An important factor underlying the effectiveness of regenerative responses to an injury is the plasticity of the stem/progenitor

cells in a tissue (*Burda and Sofroniew, 2014*). The degree to which the microenvironment and specific cell types within it provide pro- or anti-regenerative factors is highly context dependent. The neonatal mouse cerebellum has a remarkable capacity to regenerate cells ablated around birth (*Wojcinski et al., 2017*, *Bayin et al., 2021*, *Bayin et al., 2018*; *Altman and Anderson, 1971*). Thus, the cerebellum provides an ideal system to study the roles that signals in the microenvironment play in key steps of the repair process in the brain.

The cerebellum is a folded hindbrain structure that is critical for motor coordination. It also participates in higher-order social and cognitive behaviors through its circuit connections with all other brain regions (*Badura et al., 2018*; *Buckner, 2013*; *Burda and Sofroniew, 2014*; *Salman and Tsai, 2016*; *Strick et al., 2009*; *Tomlinson et al., 2013*). Compared to the rest of the brain, the cerebellum has protracted development, as its major growth occurs during the first 2 weeks after birth in mice and at least 6 months surrounding birth in humans (*Altman and Bayer, 1997*; *Rakic and Sidman, 1970*; *Dobbing and Sands, 1973*). This timing of the major growth of the cerebellum makes it susceptible to injury around birth. Indeed, cerebellar hypoplasia is the second leading risk factor for autism spectrum disorders, and cerebellar injury around birth can have devastating outcomes and significant effects on subsequent quality of life (*Tsai et al., 2018*; *Stoodley et al., 2017*; *Wang et al., 2014*). Therefore, it is critical to better understand the regenerative processes that allow repair of the cerebellum.

All the cell types in the cerebellum are derived from two progenitor zones, the embryonic rhombic lip and the ventricular zone that give rise to the excitatory neurons, or the inhibitory neurons and glia, respectively (*Leto et al., 2016*; *Joyner and Bayin, 2022*). During postnatal growth, the rhombic lip-derived granule cell precursors (GCPs) cover the surface of the cerebellum in a structure named the external granule layer (EGL) and continue to proliferate in a sonic hedgehog (SHH) dependent manner for 2 weeks after birth in mice (*Wechsler-Reya and Scott, 1999*; *McMahon et al., 2003*; *Corrales et al., 2006*). Following their exit from the cell cycle, the granule cells (GCs) migrate inward to form the internal granule layer (IGL). Other SHH-dependent progenitor populations of the neonatal cerebellum are either gliogenic Nestin-expressing progenitors (NEPs) that express SOX2 and generate astroglia (astrocytes and Bergman glia) or neurogenic-NEPs that generate late-born interneurons (*Bayin et al., 2021*, *Cerrato et al., 2018*; *Parmigiani et al., 2015*). Gliogenic-NEPs reside either in the Bergmann glia layer (BgL) intermixed with Purkinje cells and generate Bergmann glia (Bg) and astrocytes, or in the white matter (WM) in the center of the lobules (folds) and generate astrocytes. Neurogenic-NEPs are restricted to the WM and produce interneurons that migrate outward to the outermost molecular layer (*Bayin et al., 2021*, *Brown et al., 2020*; *De Luca et al., 2015*). Surprisingly, when the GCPs are killed upon injury soon after birth, the gliogenic-NEPs in the BgL (BgL-NEPs) undergo adaptive reprogramming to generate GCPs and replenish the EGL via a transitory cellular state that involves upregulation of the neurogenic gene *Ascl1* to promote a glial-to-neural fate switch (*Wojcinski et al., 2017*, *Bayin et al., 2021*). Adaptive reprogramming involves multiple sequential stages starting with increased proliferation of BgL-NEPs, then a fate switch to neuronal progenitors, migration to the site of injury (EGL), and acquisition of a GCP identity. The full repertoire of injury-induced signals that initiate and govern adaptive reprogramming remains to be discovered.

In the adult brain, numerous cell types communicate and provide a concerted response to injury, including astrocytes, microglia (macrophages of the brain), and stem cells of the neurogenic niches (*Frik et al., 2018*). The timelines of the cellular responses of each cell type to injury – cell death, activation of microglia, reactive gliosis, proliferation, scar formation, and cellular remodeling – have been delineated for specific adult brain injuries, particularly in the cerebral cortex. For example, upon traumatic brain injury, cells release damage-associated molecular patterns (DAMPs), which act as an inflammatory stimulus and activate microglia that can lead to gliosis, eventually causing neurotoxicity and scarring (*Donat et al., 2017*). However, the cellular composition and microenvironment of the early postnatal brain are very different from the adult. In the neonatal cerebellum, microglia are immature (*Li et al., 2019*) and are still being generated, and NEPs and GCPs are actively proliferating and producing astroglia and neurons. Therefore, the existing knowledge on how adult brain cells react to injury might not apply to the neonatal cerebellum. For example, in the spinal cord, while neonatal microglia and astrocytes facilitate scarless repair, the same cells in the adult promote scarring upon spinal cord injury in mice (*Li et al., 2020*). It is thus important to study the microenvironment of the neonatal brain during repair to determine what factors promote or inhibit regeneration.

Dying cells release many factors, including reactive oxygen species (ROS) that activate signaling cascades in neighboring cells. However, little is known about how these signals regulate brain repair, especially during development. The level of ROS during homeostasis is regulated by metabolic processes and typically is increased following injury (*Niethammer, 2016*). Furthermore, ROS can directly react with proteins that regulate proliferation, viability, quiescence, differentiation, and metabolism (*Bigarella et al., 2014*; *Tan and Suda, 2018*). Thus, ROS are considered key signaling molecules that participate in the crosstalk between progenitor cell fate decisions and metabolic switches in a context- and cell type-dependent manner (*Bigarella et al., 2014*). One significant mechanism by which ROS signaling is implicated during inflammatory responses following an injury is through the activation of microglia, which in turn can lead to more ROS production (*Smith et al., 2022*). This process is critical as it can potentially promote repair. However, the role of ROS signaling during adaptive reprogramming of NEPs following neonatal cerebellar injury remains unknown.

Here, we first delineate the sequential changes in the microenvironment upon injury (focused irradiation) to the mouse hindbrain at postnatal day 1 (P1). We then demonstrate a requirement for a transient increase in ROS levels at ~24 hr post-injury for cerebellar regeneration. Single-cell RNA-sequencing (scRNA-seq) and bulk assay for transposase-accessible chromatin with sequencing (ATAC-seq) analyses revealed increased ROS signaling compared to controls that peaks 24 hr after injury in NEPs, demonstrating that ROS is an acute signal associated with the NEP response to GCP death. A functional role of ROS signaling was established using a transgene (mCAT) that expresses the human mCAT which can reduce ROS levels broadly. Several key steps in adaptive reprogramming were abrogated in mCAT mice leading to reduced replenishment of the EGL and a smaller adult cerebellum. Finally, we show that the density of microglia is reduced at P5 in irradiated mCAT mice compared to controls and provide preliminary evidence that microglia play a role in the step of replenishing the EGL with BgL-derived GCPs during adaptive reprogramming.

## Materials and methods

### Animals

All the mouse experiments were performed according to protocols approved by the Institutional Animal Care and Use Committee of Memorial Sloan Kettering Cancer Center (MSKCC) (protocol no. 07-01-001). Animals were housed on a 12 hr light/dark cycle and given access to food and water ad libitum.

Two mouse lines were used in this study: Nes-Cfp (JAX #034387) (*Encinas et al., 2006*) and mCAT (JAX #016197) (*Schriner et al., 2005*). For both transgenic mouse lines, we analyzed mice that had a single transgene insertion on one chromosome. Animals were maintained on an outbred Swiss Webster background. Both sexes were used for analyses except for the genomics experiments (scRNA-seq and ATAC-seq) where males were used.

### EdU administration

5-Ethynyl-2′-deoxyuridine (EdU) stock was dissolved in sterile phosphate-buffered saline (PBS) at 10 mg/mL, and a dose of 5 µg/g was intraperitoneally injected into animals 1 hr prior to euthanasia.

### PLX5622 administration

PLX5622 powder was provided by Plexxikon under a Materials Transfer Agreement. PLX5622 powder was first diluted in DMSO at 20 mM and then diluted 1× in PBS just before intraperitoneal injection into newborn pups. Injections were given every day from P1 to P8 at a dose of 10 µg/g of body weight. Control pups were injected with PBS-DMSO vehicle control.

### Irradiation

P1 mice were anesthetized by hypothermia and given a single dose of ~5 Gy γ-irradiation in an X-RAD 225Cx (Precision X-ray) Microirradiator in the MSKCC Small-Animal Imaging Core Facility. A 5 mm diameter collimator was used to target the hindbrain from the left side of the animal.

**Table 1.** List of antibodies and related information.

| Antigen | Species | Concentration | References | Source |
| --- | --- | --- | --- | --- |
| Catalase | Rabbit | 1–100 | 01-05-030000 | Athens Research and Technology |
| GFAP | Chicken | 1–500 | ab4674 | Abcam |
| GFP | Rat | 1–1000 | 440484 | Nacalai Tesque |
| IBA1 | Rabbit | 1–500 | 019-19741 | Wako Chemicals |
| SOX2 | Goat | 1–500 | AF2018 | R&D Systems |

## Tissue preparation and histology

For immunocytochemistry, animals younger than P8 were sacrificed and then brains were dissected, fixed in 4% paraformaldehyde for 24 hr at 4°C, cryoprotected in 30% sucrose in PBS until they sank and then frozen in Cryo-OCT (Tissue-Tek). Older animals were anesthetized and then perfused with cold PBS followed by 4% paraformaldehyde prior to brain dissection. Frozen brains were cryosectioned sagittally at 14 µm and slides stored at –20°C. Midline cerebellar sections were used for quantification in all downstream analyses.

For immunofluorescence staining, slides were allowed to warm to room temperature (RT) and washed three times in PBS. Then, tissues were blocked for 1 hr with blocking buffer (5% bovine serum albumin [BSA] [wt/vol] in 1× PBS with 0.1% Triton X-100) at RT. Primary antibodies diluted in the blocking buffer were placed on slides for overnight incubation at 4°C. Slides were then washed in PBS with 0.1% Triton X-100 and incubated with fluorophore-conjugated secondary antibodies diluted in the blocking buffer for 1hour (hr) at room temperature (RT). Following washes in PBS with 0.1% Triton X-100 after the secondary antibody incubation, nuclei were counterstained with Hoechst (1:3000), and the slides were mounted using mounting media (Electron Microscopy Sciences). Primary antibodies used are described in *Table 1*, and secondary antibodies were Alexa Fluor-conjugated secondary antibodies (1:1000).

5-ethynyl-2′-deoxyuridine (EdU) was detected using a Click-it EdU (Invitrogen, C10340) assay with Sulfo-Cyanine5 azide (Lumiprobe Corporation, A3330). FOR TUNEL staining, after primary antibody incubation and washes, sections were permeabilized in PBS with 0.5% Triton X-100 for 10 minutes (min) and then preincubated in TdT buffer (30 mM Tris HCl, 140 mm sodium cacodylate, and 1 mM $CoCl_2$) for 15 min at RT. Slides were then incubated for 1 hr at 37°C in TUNEL reaction solution containing Terminal Transferase and Digoxigenin-11-dUTP (Roche). After the TUNEL reaction and washes, slides were incubated with a secondary antibody solution which included a sheep anti-dixogenin-rhodamine (Roche) for the visualization of TUNEL reaction.

## Image acquisition and analysis

Images were collected with a DM6000 Leica microscope, a NanoZoomer Digital Pathology microscope (Hamamatsu Photonics), or an LSM880 confocal microscope (Zeiss). Images were processed using NDP.view2 software, ImageJ software (NIH, Bethesda, MA, USA), and Photoshop (Adobe).

## Cell dissociation for FACS and flow cytometry

Cerebella were dissected into ice-cold 1x Hank's Buffered Salt Solution (Gibco) and dissociated using Accutase (Innovative Cell Technologies) at 37°C for 10–15 min. After dissociation, Accutase was diluted with 3× excess volume of neural stem cell media (Neurobasal medium, supplemented with N2, B27 [without vitamin A]), and nonessential amino acids (Life Technologies, Gibco) and cells were filtered using a 40 µm mesh cell strainer. After 5 min of centrifugation in a chilled centrifuge at 500×*g*, the pellet was resuspended in neural stem cell media and strained using strainer cap tubes (Falcon) for downstream experimentation. All centrifugation was performed at 4°C and cells were kept on ice when possible.

## Flow cytometry analysis for ROS and mitochondria mass

For MitoSOX and MitoTracker analyses, cells were incubated for 30 min at 4°C with 5 µM of MitoSOX and 100 µM of MitoTracker (Thermo Fisher Scientific) to assess mitochondrial superoxide production

and mitochondrial mass, respectively. Data were collected using an LSR Fortessa flow cytometer (BD) and analyzed using FlowJo software. The gating for the MitoSOX or MitoTracker high populations (top 90% intensity) was performed as previously described (*Clutton et al., 2019*).

## Multiplexed scRNA-seq

### Sample preparation

2–4 Nes-Cfp cerebella/replicates from male control non-irradiated pups or pups that were irradiated at P1 were collected at P1 (control only), P2, P3, and P5 and dissociated as described above. All conditions were performed in 2 replicates for nonIR and IR conditions, except for P5. Following dissociation, CFP+ cells were immediately sorted on a BD FACS Aria sorter (BD Biosciences) using a 100 μm nozzle. 50,000 Nes-CFP+ cells from each sample were processed for scRNA-seq. Cells were sorted into neural stem cell media.

### Multiplexing, droplet preparation, sequencing, and data processing

The scRNA-seq of fluorescence-activated cell sorting (FACS)-sorted Nes-CFP+ cell suspensions was performed on a Chromium instrument (10x Genomics) following the user guide manual for 3′ v3.1. In brief, 50,000 FACS-sorted cells from each condition were multiplexed using CellPlex reagents (10x Genomics) as described by the manufacturer's protocol. P3 nonIR replicate #2 did not yield sufficient cells after multiplexing. The viability of cells was above 95%, as confirmed with 0.2% (wt/vol) Trypan Blue staining, and barcoded cells were pooled into a single sample in PBS containing 1% BSA to a final concentration of 700–1300 cells/μL. 2–3000 cells were targeted for each sample. Samples were multiplexed together on one lane of a 10x Chromium following the 10x Genomics 3′ CellPlex Multiplexing protocol, and a total of ~30,000 cells were targeted for droplet formation. Cells were captured in droplets. After the reverse transcription and cell barcoding in droplets, emulsions were broken, and cDNA was purified using Dynabeads MyOne SILANE followed by PCR amplification per the manufacturer's instruction. Final libraries were sequenced on an Illumina NovaSeq S4 platform (R1 – 28 cycles, i7 – 8 cycles, R2 – 90 cycles) by the MSKCC Core Facility.

### Data analysis

scRNA-seq FASTQ files were demultiplexed using sharp (*Chun, 2022*) and initially mapped to the mouse mm10 reference genome using Cell Ranger v6.0.1 (*Zheng et al., 2017*). The Cell Ranger BAM files for individual samples were then converted back to FASTQ files using *bamtofastq* (Cell Ranger v7.0.1), with --reads-per-fastq=1000000000. Reads were then mapped to the GRCm39 mouse reference with GENCODE annotations (vM30) using STARsolo v2.7.10a (`--soloFeatures` Gene Velocyto, `--soloType` CB_UMI_Simple, `--soloCBwhitelist` 3M-february-2018.txt, `--soloUMIlen` 12, `--soloCellFilter` EmptyDrops_CR, `--soloMultiMappers` EM) (*Kaminow et al., 2021*; *Frankish et al., 2021*).

The STARsolo output was used to create Seurat (v4.3) objects for each sample with spliced and unspliced read count matrices (*Hao et al., 2021*). The objects were integrated by running NormalizeData and FindVariableFeatures (selection.method = 'vst', nfeatures = 3000) on each one, then SelectIntegrationFeatures and FindIntegrationAnchors on the list of objects and finally IntegrateData with the anchors. Counts were then normalized with SCTransform. Based on manual analysis, cells were filtered out where the number of detected genes was ≤1500, the number of detected transcripts was ≥40,000, and mitochondrial gene percentage ≥5%. To determine cell cycle phases, the *Kowalczyk et al., 2015*, cell cycle markers were used, assuming the gene names, capitalized to the title case, to be orthologous between mouse and human with the CellCycleScoring function. SCTransform was used to regress out the 'Cell Cycle difference' score (S score – G2M score). Dimension reduction was performed using RunPCA and RunUMAP (dims = 1:20, n.neighbors=30). For clustering, FindNeighbors was run using the first 20 principal components, then FindClusters with the Leiden algorithm with the default settings (*Traag et al., 2019*). Clusters were annotated using known markers: GCPs (*Atoh1+/Barhl1+/Tubb3+*), BgL-NEPs (*Hopx+*), ependymal cells (*Foxj1+*), immature interneurons (*Pax2+*), neurogenic-NEPs (*Ascl1+*), astrocytes (*Slc6a11+*), oligodendrocytes (*Olig1+*), meninges (*Col3a1+/Vtn+/Dcn+*), microglia (*Ly86+/Fcer1g+*).

The differential gene expression analyses were performed individually on *Hopx+* gliogenic-NEPs, *Ascl1+* neurogenic-NEPs, and GCPs following the same computational workflow. First, clusters

containing *Hopx*-NEPs (clusters 2, 3, 6, 10), *Ascl1*-NEPs (clusters 5, 8, 11), or GCPs (clusters 1, 4, 7, 12, 14) were subsetted from the original dataset based on the expression of *Hopx*, *Ascl1*, or *Atoh1*, *Barhl1*, plus *Rbfox*, respectively. Second, the subsetted cells were divided according to whether the cells were from P2 or P3+P5 and split based on their biological replicate. The split datasets were normalized using NormalizeData with default parameters, and the 3000 top variable features were computed using FindVariableFeatures with default settings. Reintegration of the datasets was subsequently performed using SelectIntegrationFeatures, FindIntegrationAnchors, and IntegrateData all with default parameters as previously described, except for IntegrateData in the *Ascl1+* neurogenic-NEP analyses where k.weight=50 was used. Following reintegration, SCTransform with default parameters was used to normalize mitochondrial read percentage, cell cycle difference score, number of features, and number of counts. Dimension reduction was thereafter performed using RunPCA with default parameters and RunUMAP with default parameters except dims = 1:40, repulsion. strength=0.1, and min.dist=0.5. Clustering was subsequently performed using FindNeighbors with default settings except dim = 30 and FindClusters with default settings for resolutions between 0.1 and 3 using the original Louvain algorithm. A final resolution that gave a high silhouette score with a relatively low negative silhouette proportion was selected for individual datasets. To allow downstream DESeq2 analyses, count matrices were constructed by aggregating counts from cells from the same treatment condition and biological replicate using AggregateExpression. The aggregated count matrices were converted into a DESeq2 dataset object using DESeqDataSetFromMatrix, grouping the samples by treatment conditions. Genes with fewer than 10 counts were filtered out. DESeq2 (v1.36.0) was used to perform the differential expression analyses using a negative binomial distribution and default settings (*Love et al., 2014*). The results were visualized using EnhancedVolcano with a fold change cutoff of ±0.5 and an adjusted p-value threshold of 0.05.

The gene ontology (GO) term analyses were performed on differentially expressed genes from the DESeq2 results filtered with a $\log_2$fold-change threshold of ±0.5 and an adjusted p-value threshold of 0.05 for each comparison. The probability weight function was computed using nullp with default parameters and the mm8 mouse genome. The background genes used to compute the GO term enrichment include all genes with gene symbol annotations within mm8. The category enrichment testing was performed using goseq with default parameters from the goseq package (v1.48.0).

## Bulk ATAC-seq

### Sample preparation

FACS-sorted Nes-CFP+ cells (30,000–50,000 per replicate) were isolated from control or irradiated (at P1) P2 cerebella. 2–3 cerebella were pooled for each sample. Cells were immediately processed for nuclei preparation and transposition using the OMNI-ATAC protocol (*Corces et al., 2017*). Sequencing was performed at the MSKCC Genomics Core Facility using the Illumina NovaSeq S4 platform.

### Data processing and analysis

Raw sequencing reads were 3' trimmed and filtered for quality and adapter content using version 0.4.5 of Trim Galore (https://www.bioinformatics.babraham.ac.uk/projects/trim_galore), with a quality setting of 15, and running version 1.15 of cutadapt and version 0.11.5 of FastQC. Version 2.3.4.1 of bowtie2 (http://bowtie-bio.sourceforge.net/bowtie2/index.shtml) was used to align reads to mouse assembly mm10 and alignments were deduplicated using MarkDuplicates in version 2.16.0 of Picard Tools. Enriched regions were discovered using MACS2 (*Liu, 2023*) with a p-value setting of 0.001, filtered for blacklisted regions (http://mitra.stanford.edu/kundaje/akundaje/release/blacklists/mm10-mouse/mm10.blacklist.bed.gz), and a peak atlas was created using ±250 bp around peak summits. The BEDTools suite (http://bedtools.readthedocs.io) was used to create normalized bigwig files. Version 1.6.1 of featureCounts (http://subread.sourceforge.net) was used to build a raw counts matrix, and DESeq2 was used to calculate differential enrichment for all pairwise contrasts. Peak-gene associations were created by assigning all intragenic peaks to that gene, while intergenic peaks were assigned using linear genomic distance to transcription start site. Network analysis was performed using the assigned genes to differential peaks by running enrichplot::cnetplot in R with default parameters. Composite and tornado plots were created using deepTools v3.3.0 by running computeMatrix and plotHeatmap on normalized bigwigs with average signal sampled in 25 bp windows and flanking

region defined by the surrounding 2 kb. Motif signatures were obtained using Homer v4.5 (http://homer.ucsd.edu) on differentially enriched peak *regions*.

## Quantification and statistical analysis

For detecting TUNEL, IBA1, GFAP, CFP, and SOX2, images were captured using a DM600 Leica fluorescent microscope and subsequently quantified on ImageJ Software (NIH). Measurements were conducted on lobules 3, 4, and 5 of midsagittal sections unless indicated in Figure legends. Positive cells were counted and densities were calculated based on BgL length, EGL area, WM area, or on the whole cerebellum without the EGL (outside EGL) as indicated in the figures. For the cerebellar section area, images were acquired using a NanoZoomer 2.0 HT slide scanner (Hamamatsu). Midsagittal sections were selected and exported for manual analysis using ImageJ software. For EGL thickness, images of lobules 3, 4, and 5 from midsagittal sections were obtained using a DM600 Leica fluorescent microscope and analyzed on ImageJ. EGL thickness was calculated as the EGL area divided by the EGL perimeter.

Prism (GraphPad) was used for all statistical analyses. Statistical tests performed in this study were Welch's two-tailed t-test and two-way analysis of variance (ANOVA) followed by post hoc analysis with Tukey's multiple comparison tests. A p-value≤0.05 was considered significant. Results are presented as the mean ± SEM. At least three biological samples and two to three sections per sample were analyzed for each experiment to ensure reproducibility, and the sample sizes are reported in the Results section for significant data. For qualitative analysis, midsagittal sections from at least four samples were observed.

## Results

### scRNA-seq of NEPs during adaptive reprogramming reveals increased cellular stress, ROS signaling, and DNA damage

To investigate the molecular changes that NEP subpopulations undergo upon injury to the EGL, in particular an increase in the signaling pathways associated with injury-induced cellular stress and ROS, we performed multiplexed scRNA-seq of CFP+ cells isolated by FACS of cerebella from Nes-Cfp transgenic neonates that express CFP from a Nestin gene promoter and enhancer either irradiated at P1 (IR; P2, P3, P5) or non-irradiated (nonIR; P1, P2, P3, P5) (*Figure 1A*, *Figure 1—figure supplement 1A and B*). Following the filtering out of poor quality cells and integration of replicates and the two conditions, the clustering of 11,878 cells (6978 nonIR and 4900 IR) was performed (*Hao et al., 2021*). The analysis revealed the expected three distinct groups of cells: gliogenic-NEPs and astrocytes (*Hopx+* clusters 2, 3, 6, 10), neurogenic-NEPs (*Ascl1+* clusters 5, 8, 11), and GCP (*Atoh1+* clusters 1, 4, 7, 12, 14) that were present at each stage and in both conditions (*Figure 1B–E*, *Figure 1—figure supplement 1C–F*, *Figure 1—source data 1*). These groups of cells were further subdivided into molecularly distinct clusters based on marker genes and their cell cycle profiles or developmental stages (*Figure 1D*, *Figure 1—source data 1*). In addition, oligodendrocyte progenitors (cluster 15), microglia (clusters 17, 21), ependymal cells (clusters 13, 18, 19), and meninges (cluster 16) were detected (*Figure 1B and D*, *Figure 1—source data 1*). Cluster 20 represented low-quality cells and was omitted from downstream analyses. As expected, the GCP clusters were enriched in the cells from irradiated mice and at P5 (*Figure 1—figure supplement 1C*). Detection of *Nes* mRNA confirmed that the transgene reflects endogenous *Nes* expression in progenitors of many lineages, and also that the perdurance of CFP protein in immediate progeny of *Nes*-expressing cells allowed the isolation of these cells by FACS (*Figure 1E*).

Our further analyses focused on changes in the signaling pathways associated with injury-induced cellular stress and ROS. We previously showed that a subset of the *Hopx+* gliogenic-NEPs that are present in the BgL are the ones that undergo adaptive reprogramming following GCP death (*Wojcinski et al., 2017*, *Bayin et al., 2021*). We therefore assessed the immediate and later effects of GCP death on *Hopx+* gliogenic-NEPs by performing differential expression analyses between nonIR and IR gliogenic-NEPs (*Hopx+*, clusters 2, 3, 6, 10) at P2, or at P3 and P5 (P3+5). 24 hr after injury at P1, 132 genes in gliogenic-NEP clusters were significantly upregulated in IR compared to 34 genes that were upregulated in nonIR P2 cells (adjusted p-value≤0.05, *Figure 1F*, *Figure 1—source data 2*). The significantly increased genes included *Cdkn1a*, *Phlda3*, *Ass1*, and *Bax*, all of which have been

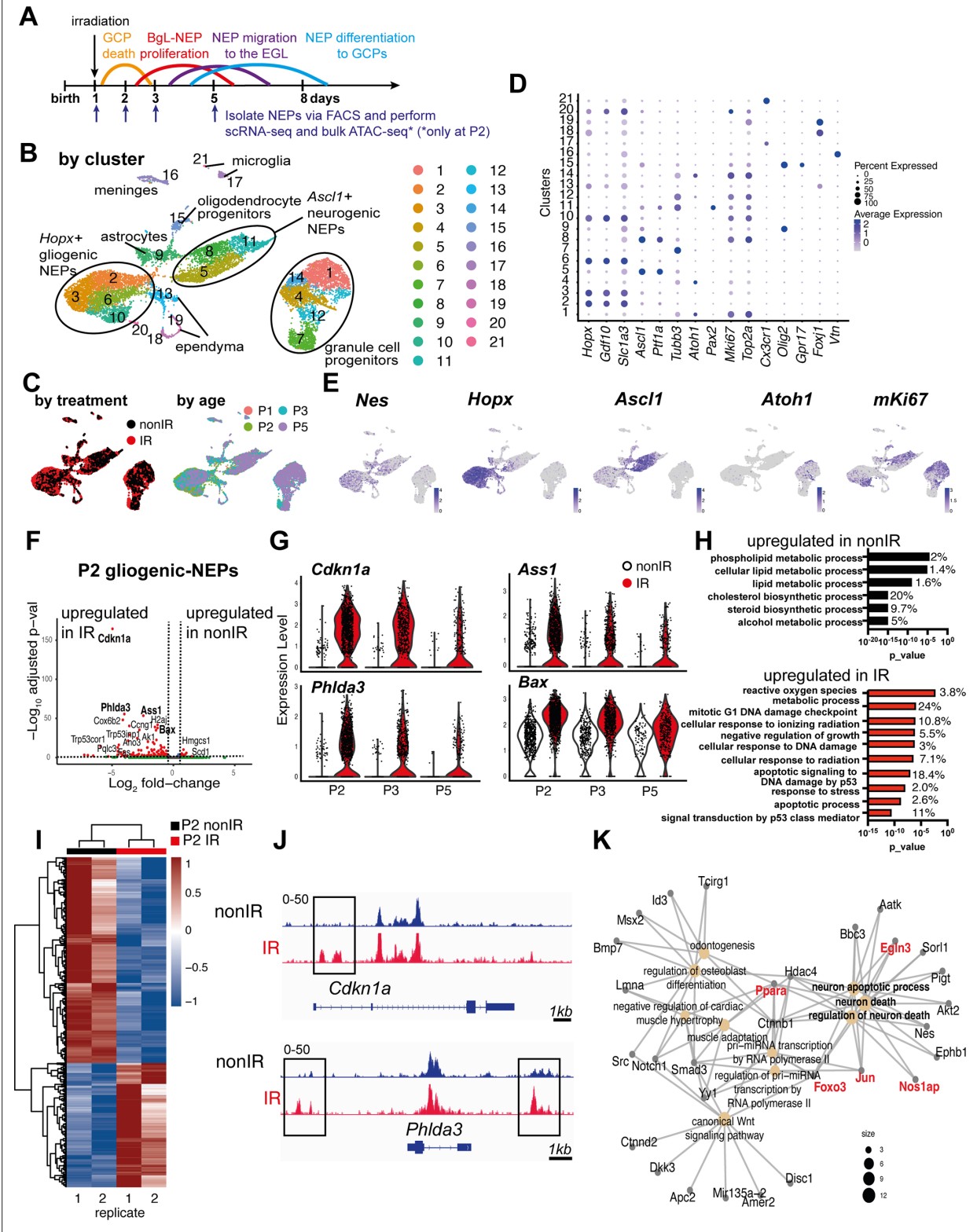

**Figure 1.** Injury induces reactive oxygen species (ROS) and cell stress signaling reflected by changes in the transcriptome and chromatin landscape of progenitors. (**A**) Schematic summarizing the experimental plan. (**B–C**) UMAPs of 11,878 cells (6978 nonIR and 4900 IR) showing cluster annotations (**B**), treatment (black: nonIR, red: IR), and the age of the samples (red: P1, green: P2, blue: P3, purple: P5) (**C**). (**D**) Dot plot showing the expression levels of key marker genes used for cluster annotation (gliogenic-Nestin-expressing progenitors [NEPs]: *Hopx, Gdf10, Slc1a3*, neurogenic-NEPs: *Ascl1, Ptf1a*, immature neurons: *Pax2*, granule cell precursors [GCPs]: *Atoh1*, postmitotic neurons: *Tubb3*, microglia: *Cx3cr1*, oligodendrocyte progenitors: *Olig2*,

*Figure 1 continued on next page*

*Figure 1 continued*

oligodendrocytes: *Gpr17*, ependymal cells: *Foxj1*). (**E**) Feature plots showing *Nes*, *Hopx* (gliogenic-NEPs), *Ascl1* (neurogenic-NEPs), *Atoh1* (GCPs), and *mKi67* (proliferation) expression highlighting the three main populations of interest. Clusters containing *Hopx*-NEPs (clusters 2, 3, 6, 10), *Ascl1*-NEPs (clusters 5, 8, 11), or GCPs (clusters 1, 4, 7, 12, 14) were subsetted from the original dataset and were divided according to age (P2 or P3+P5) for the downstream differential expression analyses. (**F**) Volcano plot showing differentially expressed genes in the P2 gliogenic-NEPs (red: adjusted p-value≤0.05, log$_2$fold-change>|1|). (**G**) Violin plots showing some of the top differentially expressed genes in P2 gliogenic-NEPs and how their expression changes over time with respect to their expression in control cells. (**H**) Some of the significant gene ontology (GO) terms associated with differentially expressed genes in P2 gliogenic-NEPs that were either upregulated in nonIR (top panel) or IR (bottom panel) cells (adjusted p-value≤0.05, *Figure 1—source data 2*). (**I**) Heatmap showing differentially open chromatin regions in P2 nonIR and IR NEPs, identified by bulk assay for transposase-accessible chromatin with sequencing (ATAC-seq) (1168 differentially open regions, adjusted p-value<0.05, *Figure 1—source data 3*). (**J**) Tracks highlighting the injury-induced open chromatin regions around *Cdkn1a* and *Phlda3*, the top differentially expressed genes identified in (**F**). (**K**) Linkages between genes and GO terms identified by the ATAC-seq data revealed an active transcriptional network involved in regulating cell death and apoptosis. Genes colored in red (*Ppara*, *Egln3*, *Foxo3*, *Jun*, and *Nos1ap*) have been implicated as upregulated with increased ROS levels or involved in ROS signaling.

The online version of this article includes the following source data and figure supplement(s) for figure 1:

**Source data 1.** Marker genes and all genes expressed by cluster in single-cell RNA-sequencing (scRNA-seq) dataset for irradiated at P1 (IR; P2, P3, P5) or non-irradiated (nonIR; P1, P2, P3, P5).

**Source data 2.** Pseudobulk differential expression analysis between nonIR and IR gliogenic-Nestin-expressing progenitors (NEPs) (*Hopx+*, clusters 2, 3, 6, 10), neurogenic-NEPs (*Ascl1+*, clusters 5, 8, 11), and granule cell precursors (GCPs) (*Atoh1+*, clusters 1, 4, 7, 12, 14) at P2, or at P3 and P5 (P3+5).

**Source data 3.** Gene ontology (GO) term analyses of differentially expressed genes (*Figure 1—source data 2*) of nonIR and IR gliogenic-Nestin-expressing progenitors (NEPs) (*Hopx+*, clusters 2, 3, 6, 10), neurogenic-NEPs (*Ascl1+*, clusters 5, 8, 11), and granule cell precursors (GCPs) (*Atoh1+*, clusters 1, 4, 7, 12, 14) at P2, or at P3 and P5 (P3+5).

**Source data 4.** Differentially open peaks at P2 identified by bulk assay for transposase-accessible chromatin with sequencing (ATAC-seq) from nonIR and IR Nestin-expressing progenitors (NEPs).

**Source data 5.** Motif analysis of regions with increased accessibility in IR Nestin-expressing progenitors (NEPs) compared to the nonIR at P2.

**Figure supplement 1.** Single-cell RNA-sequencing (scRNA-seq) quality metrics and number of cells sequenced in each condition and biological replicate.

**Figure supplement 2.** Injury induces distinct transcriptional changes in Nestin-expressing progenitor (NEP) subtypes and granule cell precursors (GCPs) during adaptive reprogramming.

implicated as increased in response to DNA damage and in ROS signaling, as well as in anti-apoptotic functions (*Figure 1G*; *Masgras et al., 2012*; *Bensellam et al., 2019*; *Qiu et al., 2014*; *Jiang et al., 2008*). Indeed, many of the significant GO terms associated with the genes upregulated in gliogenic-NEPs with injury were related to response to irradiation, DNA damage, the p53 pathway, and ROS metabolic processes, whereas many of the significant GO terms associated with the genes upregulated in the nonIR cells at P2 were related to metabolic processes (p-value≤0.05, *Figure 1H*, *Figure 1—source data 3*). Interestingly, the transcriptional changes observed at P2 were less pronounced at later time points, where although some of the top differentially expressed genes at P2 were still significantly upregulated at P3+5 stages combined (e.g. *Cdkn1a*, *Phlda3*, *Ass1*, *Bax*), the increase in expression levels of these genes upon injury and/or the number of cells expressing them gradually declined after P2 in IR gliogenic-NEPs when compared to their nonIR counterparts (*Figure 1G*). Genes upregulated in IR P3+5 gliogenic-NEPs were associated with similar GO terms to those at P2, such as response to irradiation and the p53 pathway; however, 'ROS metabolic processes' was no longer a significantly enriched GO term (p-value≤0.05, *Figure 1—figure supplement 2A and F*, *Figure 1—source data 2* and *Figure 1—source data 3*).

To further assess the injury-induced transcriptional signatures, we performed the same differential expression analysis on nonIR and IR neurogenic-NEPs (*Ascl1+*, clusters 5, 8, 11) and GCPs (*Atoh1+*, clusters 1, 4, 7, 12, 14) at P2, or at P3+P5 to identify the immediate and later changes upon injury at birth. Some of the DNA damage and apoptosis-related genes that were upregulated in IR gliogenic-NEPs (*Cdkn1a*, *Phlda3*, *Bax*) were also upregulated in the IR neurogenic-NEPs and GCPs at P2 (*Figure 1—figure supplement 2B–E*). Similar to the gliogenic-NEPs, P2 IR neurogenic-NEPs showed significant upregulation of genes associated with GO terms related to stress response and apoptosis following injury, although the ROS-related GO term was not as prominent (*Figure 1—figure supplement 2B and G*, *Figure 1—source data 2* and *Figure 1—source data 3*). Interestingly, although IR neurogenic-NEPs at P3+5 had only 27 genes that were significantly upregulated following injury (adjusted p-value≤0.05, *Figure 1—source data 2*, *Figure 1—figure supplement 2C*),

the genes included ones associated with neural stem cells and BgL-NEPs (e.g. *Id1*, *Apoe*, *Ednrb*). The latter genes could represent the transitory *Ascl1+* BgL-NEP population that induces *Ascl1* expression during adaptive reprogramming and would be included in the *Ascl1+* neurogenic clusters or reflect the delayed neurogenesis of neurogenic-NEPs previously demonstrated (*Bayin et al., 2021*). Consistent with the latter, the nonIR P3+5 neurogenic-NEPs showed an increase in mature neuron markers compared to IR cells (adjusted p-value<0.05, *Figure 1—figure supplement 2C and H*, *Figure 1—source data 2* and *Figure 1—source data 3*).

In contrast to the gliogenic- and neurogenic-NEP subtypes, P2 IR GCPs showed upregulation of genes enriched in GO terms such as neural differentiation and axonogenesis, as well as ROS signaling, and nonIR P2 GCPs showed increased expression of genes involved in cell cycle and mitosis (*Figure 1—figure supplement 2D and I*). This result could reflect the death of highly proliferative GCPs after irradiation and sparing of only postmitotic GCs upon irradiation at P1. P3+5 IR GCPs showed increased expression of genes associated with BgL-NEPs (*Id1* and *Gdf10*, adjusted p-value≤0.05) and genes associated with GO terms such as cell cycle and cell fate commitment, whereas P3+5 nonIR GCPs showed enrichment for GO terms related to cell migration and neurogenesis (adjusted p-value≤0.05, *Figure 1—figure supplement 2D, E, I, and J*, *Figure 1—source data 2* and *Figure 1—source data 3*). This result could reflect the delayed neurogenesis of GCPs following injury. Interestingly, we did not observe significant enrichment for GO terms associated with cellular stress response in the GCPs that survived the irradiation compared to controls, despite significant enrichment for ROS signaling-related GO terms (*Figure 1—source data 3*). Collectively, these results indicate that injury induces significant and overlapping transcriptional changes in NEPs and GCPs. The gliogenic- and neurogenic-NEP subtypes transiently upregulate stress response genes upon GCP death, and an overall increase in ROS signaling is observed in the injured cerebella.

## Injury induces changes in NEP chromatin landscape at P2

We next tested whether GCP death at birth induces changes to the chromatin landscape of NEPs that reflect the altered gene expression observed with scRNA-seq, by performing bulk ATAC-seq on FACS-isolated CFP+ cells from P2 control and injured Nes-Cfp pup cerebella (*Figure 1A*). P2 was chosen as it is the stage when GCPs contribute the least to the total Nes-CFP+ FACS population and to identify the immediate effects of the injury on NEP chromatin. Analysis of differentially open chromatin showed that injury induces major changes to the chromatin landscape of the NEPs (1168 differentially open regions, adjusted p-value<0.05, *Figure 1I*, *Figure 1—source data 4*). Of interest, *Cdkn1a* and *Phlda3*, two genes stimulated by ROS and injury (*Bensellam et al., 2019*; *Masgras et al., 2012*) and that were upregulated in gliogenic-NEPs after irradiation (*Figure 1F-G*, *Figure 1—figure supplement 2B*), exhibited new accessible regions around their gene bodies compared to the nonIR P2 Nes-CFP+ cells (*Figure 1J*). However, not all genes in the accessible areas were differentially expressed in the scRNA-seq data. While some of this could be due to the detection limits of scRNA-seq, further analysis is required to assess the mechanisms of how the differentially accessible chromatin affects transcription. Known motif analysis in the regions with increased accessibility upon injury showed enrichment for binding motifs of the JUN/AP1 transcriptional complex (p-value = $10^{-14}$, % of target sequences with motif = 15%, *Figure 1—source data 5*), which is known to act in response to cellular stress and be activated by ROS. In addition, the DNA binding site for FOXO3, a transcription factor that regulates ROS levels, had increased chromatin accessibility (p-value = 0.001, % of target sequences with motif = 22.41%) (*Figure 1—source data 5*; *Filosto et al., 2003*; *Auten and Davis, 2009*; *Hagenbuchner et al., 2012*). Finally, linkages between the genes in differentially open regions identified by ATAC-seq and the associated GO terms revealed an active transcriptional network involved in regulating cell death and apoptosis (*Figure 1K*). Furthermore, some of the genes involved in this response, such as *Ppara*, *Egln3*, *Foxo3*, *Jun*, and *Nos1ap*, have been implicated as upregulated with increased ROS levels or involved in ROS signaling (*Devchand et al., 2004*; *Kaelin, 2005*; *Hagenbuchner et al., 2012*; *Filosto et al., 2003*). In summary, our ATAC-seq data analysis along with the scRNA-seq provides strong evidence that irradiation causes increased ROS signaling in the BgL-NEPs upon GCP death by inducing transcriptional and epigenetic changes within 1 day after injury (P2). In addition, genes related to cell survival and death, cellular stress, and DNA damage are upregulated in NEPs shortly upon injury, possibly as a means to overcome the cellular effects of injury and induce adaptive reprogramming of NEPs to replenish the lost cells.

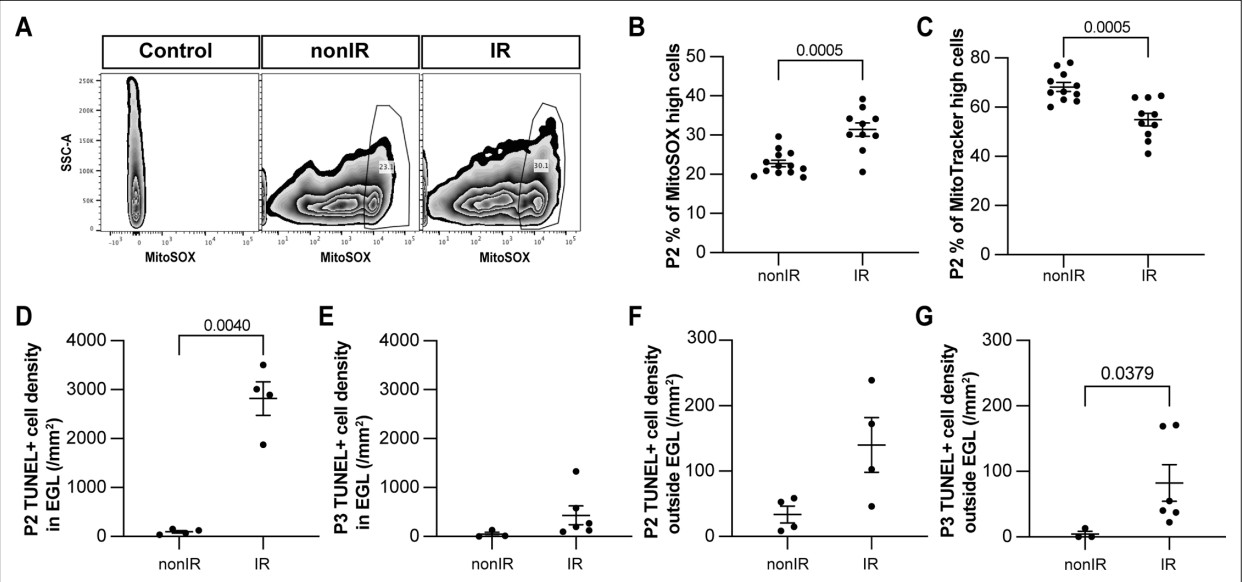

**Figure 2.** Cerebellar injury at P1 results in increased superoxide production, a reduction in mitochondria, and increased cell death in the external granule layer (EGL) that peaks 24 hr after injury. (**A**) Examples of flow cytometry analysis of mitochondrial reactive oxygen species (ROS) at P2 from nonIR and IR cerebella using MitoSOX dye. Gating determined the top 90% MitoSOX signal (MitoSOX high cells). (**B, C**) Quantification of MitoSOX high (**B**) and MitoTracker high (**C**) expression in nonIR and IR cerebella at P2. (**D, E**) Quantification of TUNEL+ cell density in the EGL at P2 (**D**) and P3 (**E**) in lobules 3–5 of nonIR and IR mice. (**F, G**) Quantification of TUNEL+ cell density outside the EGL at P2 (**F**) and P3 (**G**) in lobules 3–5 of nonIR and IR mice. EGL, external granular layer; SSC, side scatter; P, postnatal day; nonIR, non-irradiated; IR, irradiated. All statistical significance was determined using an unpaired t-test, and data are represented as mean ± SEM.

The online version of this article includes the following figure supplement(s) for figure 2:

**Figure supplement 1.** Irradiation of cerebella at P1 results in increased superoxide production and cell death and recruitment of microglia to the EGL that peaks at 24 hr.

## Transient increase in cerebellar ROS during apoptosis of GCPs

To validate that the transient increase in expression of genes associated with cellular stress and ROS signaling is due to an increase in ROS upon cerebellar injury, we quantified ROS levels in whole cerebellum samples using a mitochondrial superoxide indicator (MitoSOX) via flow cytometry. A significant increase in ROS (cells present in the top 90% MitoSOX+ intensity) was observed specifically at P2 (p=0.0005, n≥10) and not at P3 or P5 in IR pups compared to nonIR (*Figure 2A and B*, *Figure 2—figure supplement 1A and B*). Furthermore, quantification of mitochondrial mass with MitoTracker revealed a reduction only at P2 (p=0.0005, n≥10) in IR pups (*Figure 2C*, *Figure 2—figure supplement 1C–E*). Additionally, quantification of TUNEL staining in midline sagittal sections of the cerebella showed a large increase in cell death in the EGL of injured cerebella at P2 (p=0.0040, n=4), but not at P3 compared to the controls (*Figure 2D and E*; see also *Figure 3K and L*). Outside the EGL, there was a small increase in cell death after injury that was only significant at P3 (p=0.0379, n≥3) (*Figure 2F and G*). Thus, a transient increase in ROS in the cerebellum 1 day after irradiation at P1 correlates with the timing of the major death of GCPs.

## Altered glial microenvironment following death of GCPs

Given the potential importance of glial cells to regenerative cellular responses after injury, we next asked whether the glial microenvironment of the cerebellum changes during early postnatal development in response to irradiation at P1. We first analyzed the astrocyte marker GFAP, since it is generally upregulated soon after brain injury (*Burda and Sofroniew, 2014*). Most astrocytes, including the specialized Bg, express GFAP in the adult cerebellum, but the cells are generated by gliogenic-NEPs during the first 2 weeks after birth in mice. We therefore determined the normal location and timing of initiation of GFAP expression in these cells during postnatal cerebellum development. GFAP was first detected in rare astrocytes at P2 located in the WM below the lobules, with strong expression in all WM astrocytes at P5 and later stages (P8 and P12) (*Figure 3A–D*, observed in n=4 mice/stage). In

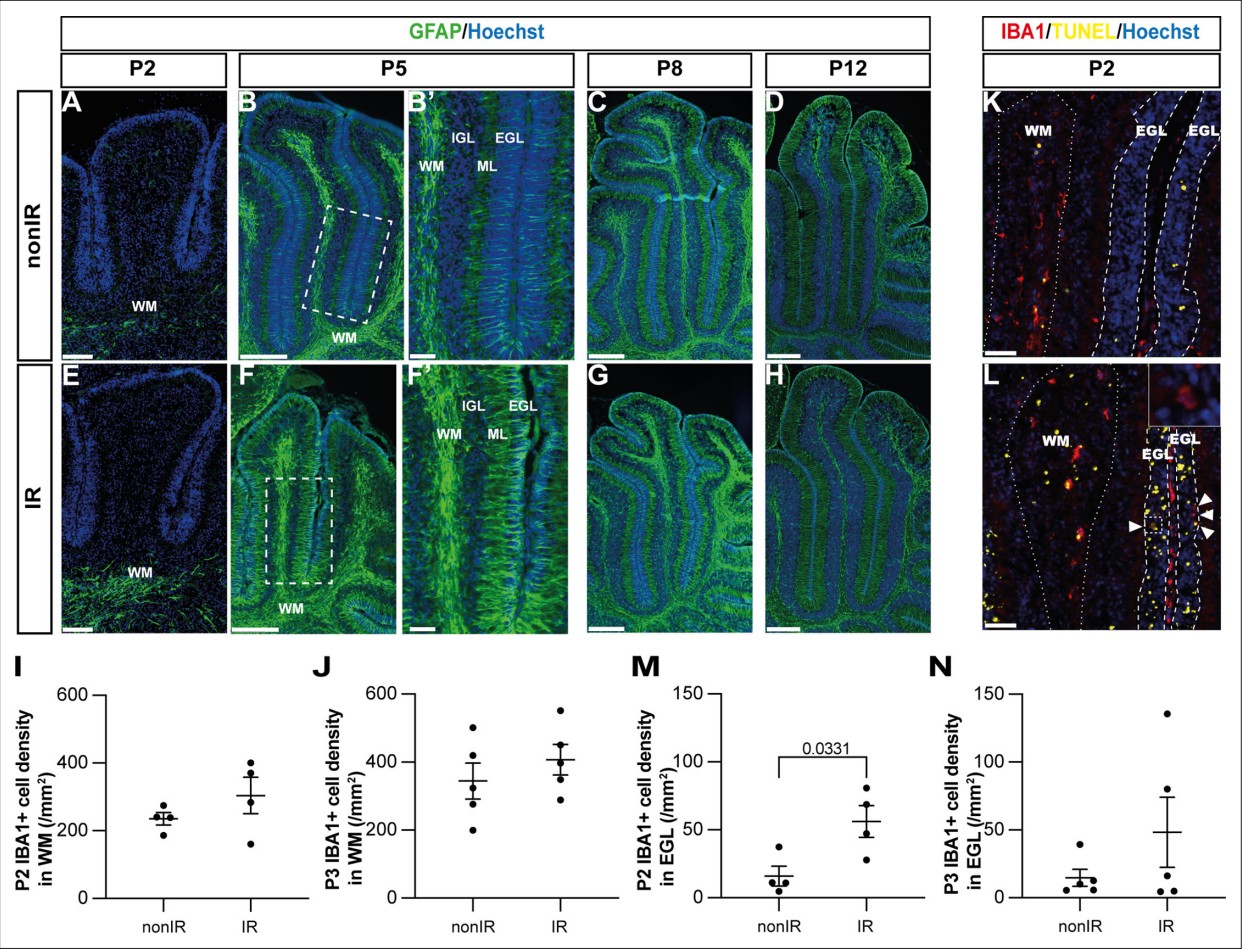

**Figure 3.** Cerebellar injury at P1 induces transient microglial recruitment to the external granule layer (EGL) and prolonged astroglial microenvironment changes in the cerebellum. (A–H) Immunohistochemical (IHC) staining of medial sagittal cerebellar sections for GFAP (green) in lobule 4/5 of nonIR and IR cerebellum at the stages indicated. Nuclei were counterstained with Hoechst. (B') and (F') show high-power images of white dashed line boxes in (B) and (F), respectively. (I, J) Quantification of IBA1+ cell density in the WM at P2 (I) and P3 (J) in lobules 3–5 of nonIR and IR mice. (K, L) IHC staining of medial sagittal cerebellar sections for IBA1 and TUNEL in lobule 3 of nonIR and IR cerebellum at P2. Nuclei were counterstained with Hoechst. WM and EGL are delineated by white dotted lines and dashed lines, respectively. High-power image in (L) of the area indicated by the white dashed line represents an IBA1+ cell present in the EGL. White arrowheads indicate additional IBA1+ cells in the EGL. (M, N) Quantification of IBA1+ cell density in the EGL at P2 (M) and P3 (N) in lobules 3–5 of nonIR and IR mice. EGL, external granular layer; WM, white matter; P, postnatal day; nonIR, non-irradiated; IR, irradiated. Scale bar: A and E: 100 µm, B, C, D, E, F, G, and H: 250 µm, B' and F': 50 µm, I and J: 50 µm. All statistical significance was determined using an unpaired t-test, and data are represented as mean ± SEM.

contrast, GFAP expression in astrocytes in the developing IGL and in the glial processes of Bg that project through the molecular layer (ML) and EGL was not detectable until P5 and was much stronger at P8 and P12 (*Figure 3A–D*). Interestingly, after injury at birth, GFAP expression was prematurely upregulated in the WM astrocytes below the lobules at P2 and in the remaining astrocytes at P5, including in the fibers of Bg that extend through the ML and EGL, compared to nonIR cerebella (*Figure 3A–H*, observed in n=4 mice/stage). GFAP expression was similar in all glia in both conditions at P8 and P12. These results reveal that astrocytes and Bg react to EGL injury caused by irradiation at P1 by initiating GFAP expression earlier than normal in the deep WM, and then around the IGL and in Bg.

The macrophages of the brain, microglia, are generated in the early embryo but their main increase in cell number occurs during neonatal development in mice (*Hammond et al., 2018*). We found that nearly all microglia were located in the WM of the cerebellum, and that the density of IBA1+ microglia in the WM increased between P2 (235.1±18.3 cells/mm$^2$) and P3 (344.3±53.1 cells/mm$^2$) and then was maintained at P5 (350.4±40.2 cells/mm$^2$) (*Figure 3I–K, Figure 2—figure supplement 1F*). Since the

area of the cerebellum is increasing between P2 and P5, active microglia production and/or infiltration must continue to occur between P2 and P5. As expected, after irradiation, the density of microglia in the injured EGL was significantly increased (~3-fold) at P2 but not at P3 or P5 compared to controls (p=0.0331, n=4) (*Figure 3L–N*, *Figure 2—figure supplement 1G*). No difference in the microglial density in the WM was detected between conditions at both time points (*Figure 3I–L*; *Figure 2—figure supplement 1F*). Thus, as expected, microglia density was increased in the EGL 1 day after injury when the maximum GCP cell death occurs.

## Decreasing ROS impairs cerebellar repair

Given the transient increase in ROS signaling in gliogenic-NEPs during peak GCP death and recruitment of microglia to the EGL, and the later astroglial response, we tested whether an increase in ROS is necessary for adaptive reprogramming and cerebellar repair following EGL injury at P1. To reduce the level of ROS, we utilized an mCAT transgenic mouse line that contains an insertion of sequences that express the human mCAT in most cells from a *CMV* promoter (*Schriner et al., 2005*). The transgene is expected to reduce mitochondrial ROS levels in all cells by catalyzing the breakdown of hydrogen peroxide into water and oxygen, hence protecting the cells from oxidative damage. We confirmed that mCAT protein is expressed throughout the cerebellum using immunohistochemical (IHC) staining of cerebellar sections (*Figure 4A and B*). MitoSOX flow cytometry revealed a significant decrease in ROS in the cerebella of mCAT mice 1 day after irradiation (P2) (p=0.0163, n≥8) and not at P3 or P5 compared to control IR mice and no baseline decrease in ROS at any stage in nonIR mice (*Figure 4C*, *Figure 4—figure supplement 1A and B*). Mitochondrial mass, as measured by MitoTracker flow cytometry, was reduced at P2 in mCAT IR compared to nonIR mCAT pups, with no significant difference observed between mCAT and control IR mice (*Figure 4D*, *Figure 4—figure supplement 1D*). The level of cell death in the EGL and density of IBA1+ microglia in the EGL and elsewhere in the cerebellum were similar between mCAT and littermate control IR mice at P2 and P3 (*Figure 4E and F*, *Figure 4—figure supplement 1G and J*). There was a slight increase in cell death outside the EGL at P2 in IR mCAT compared to nonIR mCAT mice but not compared to IR controls (*Figure 4—figure supplement 1F and E*). Thus, the mCAT transgene counteracts the transient increase in cerebellar ROS following EGL injury but does have a major effect on GCP death or the infiltration of microglia to the injured EGL (*Figure 4—figure supplement 1F–J*).

We next determined the regenerative efficiency of mCAT mice by analyzing the area of sections of cerebella from nonIR and IR P30 mCAT mice compared to littermate controls. Strikingly, the cross-sectional area of the medial cerebellum (vermis) of IR mCAT adult mice was significantly reduced compared to IR controls (p=0.0040, n≥6) (*Figure 4G–K*). Analysis of cerebellar area across ages (P3, 5, 8, and 12) revealed that the vermis sectional area of the IR mCAT cerebella was only significantly reduced at P30 compared to IR controls; however, at P12, it was reduced in mCAT IR cerebella compared to mCAT nonIR mice, whereas it was not significantly different between control IR and control nonIR mice (*Figure 4L*, *Figure 4—figure supplement 1K–N*). These results indicate that cerebellar growth begins to be reduced at P12 in mCAT mice following injury. Thus, a reduction in ROS at the time of cell death in the EGL leads to a diminution of cerebellar recovery.

## Reduced regeneration in *mCAT* mice is associated with reduced adaptive reprogramming at P5

Given that a decrease in ROS following EGL injury reduces regeneration of the neonatal cerebellum, we determined whether specific stages of the adaptive reprogramming process are altered in mCAT mice compared to controls. First, we analyzed the replenishment of the EGL by BgL-NEPs in vermis lobules 3–5, since our previous work showed that these lobules have a prominent defect. Interestingly, we found that although the thickness of the EGL in IR mice of both genotypes was similarly reduced compared to nonIR mice at P5 (p=0.0041 control and p=0.0005 mCAT; n≥4) by P8, the control EGL was a similar thickness to the control nonIR, whereas the thickness of the mCAT IR EGL was significantly reduced compared to mCAT nonIR mice (p=0.035, n=4) (*Figure 5A and B*). A key regenerative process that contributes to the expansion of the EGL following injury is the migration of BgL-NEPs to the EGL. Strikingly, the density of CFP+ cells in the EGL (mainly BgL-derived NEPs) was significantly decreased in mCAT IR mice compared to control IR mice at P5 (p=0.0002, n=4) but not P3 (*Figure 5C–G*, *Figure 5—figure supplement 1A*). Furthermore, the density of NEPs (CFP+ or

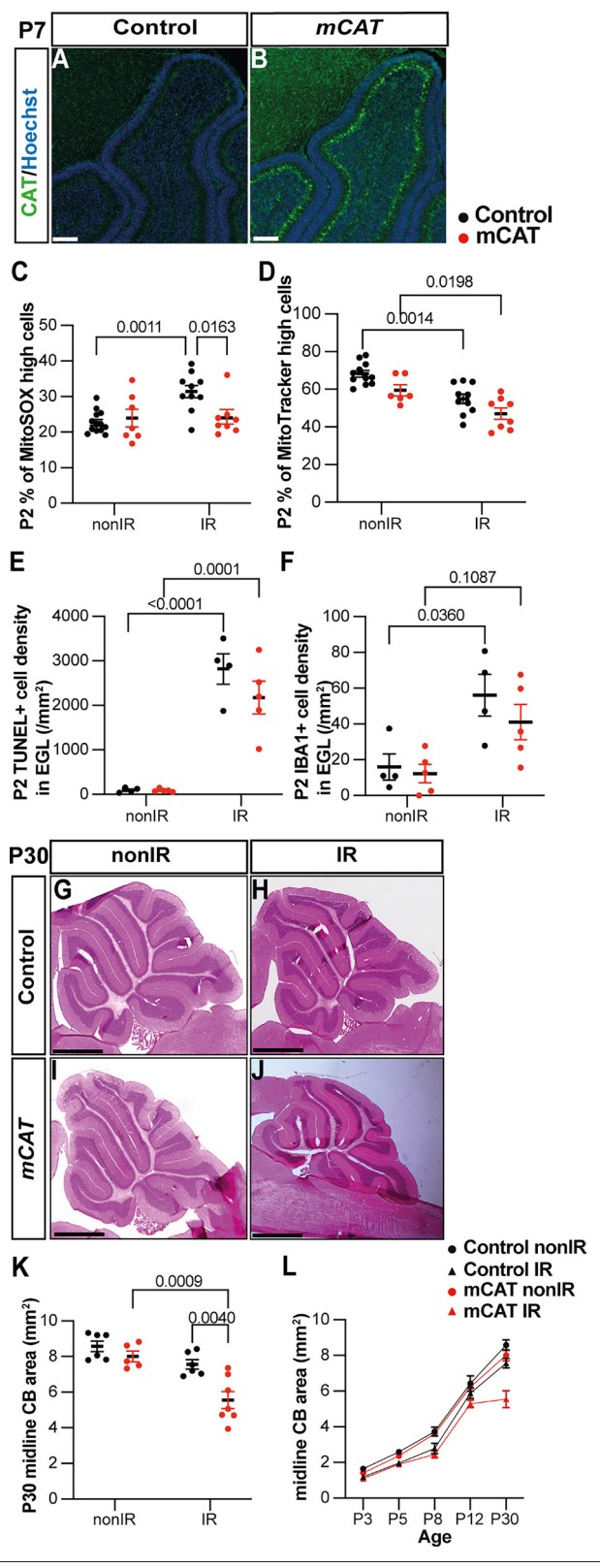

**Figure 4.** Reduction of reactive oxygen species (ROS) impairs adaptive reprogramming and cerebellar repair. (**A, B**) Immunohistochemical (IHC) staining of medial sagittal cerebellar sections for human catalase in control (**A**) and mitochondrial catalase (mCAT) mice (**B**) at P7. Nuclei were counterstained with Hoechst (blue). Similar staining was seen in four mCAT mice. (**C**) Quantification of MitoSOX high expression at P2 in control and mCAT cerebella,

*Figure 4 continued on next page*

*Figure 4 continued*

with and without irradiation at P1 (two-way ANOVA, $F_{(1,34)}=6.768$, p=0.0136). (**D**) Quantification of MitoTracker high expression at P2 in control and mCAT cerebella, with and without irradiation at P1 (two-way ANOVA, $F_{(1,31)}=25.06$, p<0.0001). (**E**) Quantification of TUNEL+ cell density in the EGL at P2 in control and mCAT cerebella, with and without irradiation at P1 (two-way ANOVA, $F_{(1,14)}=87.56$, p<0.0001). (**F**) Quantification of IBA1+ cell density in the EGL at P2 in control and mCAT cerebella, with and without irradiation at P1 (two-way ANOVA, $F_{(1,14)}=15.58$, p=0.0015). (**G–J**) Hematoxylin and eosin staining on midsagittal sections of P30 control and mCAT cerebellum with or without irradiation. (**K**) Quantification of P30 cerebellar midsagittal section area in controls and mCAT nonIR and IR mice (two-way ANOVA, $F_{(1,20)}=11.82$, p=0.0026). (**L**) Graph showing the average area of midsagittal cerebellar sections at P3, P5, P8, P12, and P30 in control and mCAT non-irradiated and irradiated mice. Detailed statistics are shown in *Figure 4—figure supplement 1*. EGL, external granular layer; P, postnatal day; nonIR, non-irradiated; IR, irradiated. Scale bar: A and B: 100 μm, F–I: 1 mm. Significant Tukey's post hoc multiple comparison tests are shown in the figures, and data are represented as mean ± SEM.

The online version of this article includes the following figure supplement(s) for figure 4:

**Figure supplement 1.** Reduction of reactive oxygen species (ROS) impairs adaptive reprogramming and cerebellar repair.

SOX2+ cells) in the BgL was significantly decreased at P5 in mCAT IR cerebella compared to controls (p=0.0010, n≥5) but not at other stages (*Figure 5H*, *Figure 5—figure supplement 1B–D*). These results indicate that BgL-NEPs have a blunted response to EGL injury and therefore do not fully expand and contribute to the replenishment of GCPs in the EGL after irradiation.

## Microglia likely contribute to one aspect of adaptive reprogramming

Given that several steps in adaptive reprogramming were decreased specifically at P5 in mCAT IR cerebella, we asked whether microglia/macrophages could be involved in any of the processes. We first determined the density of IBA1+ cells in the EGL and WM of vermis lobules 3–5 at P5 in nonIR and IR mice of both genotypes. As expected, the density of microglia in the EGL was very low in the nonIR control and mCAT cerebella (*Figure 6A–E*). Interestingly, whereas control IR mice had a similar number of IBA1+ cells in the WM as nonIR mice of both genotypes at P5, the mCAT IR mice had a lower density of microglia/macrophages in the WM compared to control IR mice at P5 (p=0.0012, n≥3) (*Figure 6A–D and F*), but no significant changes were observed at P8 (*Figure 6—figure supplement 1A*). This result raised the question of whether macrophages/microglia play a role in adaptive reprogramming.

We, therefore, tested whether reducing the density of IBA1+ microglia/macrophages after birth would alter adaptive reprogramming at P5 or cerebellar regeneration at later stages. Since macrophages and cerebellar microglia are dependent on colony stimulating factor 1 (CSF1) for their survival, we administered PLX5622, a small molecule inhibitor of CSF receptor 1 (CSFR1), to pups every day from P0 to P5 (PLX treatment) (*Kana et al., 2019*; *Tan et al., 2021*). As expected, IBA1+ cells were significantly decreased in the cerebellum of PLX-treated mice at P5 compared to their controls, both nonIR and IR (p=0.0015 and p=0.0059, n=3 and n=5, respectively) (*Figure 6G*, *Figure 6—figure supplement 1B-E*). The thickness of the EGL was not significantly altered at P5 in PLX-treated IR mice compared to IR controls (*Figure 6H*). Interestingly, similar to mCAT mice, the density of Nes-CFP+ cells in the EGL was significantly decreased in PLX-treated IR mice compared to IR controls at P5 (p=0.0008, n=5) (*Figure 6I–M*). In contrast, the density of SOX2+ cells in the BgL, corresponding to the gliogenic BgL-NEPs, was unchanged in the PLX-treated and control mice, whether irradiated or not, suggesting that the decrease in expansion of BgL-NEPs caused by ROS is not mediated by microglia (*Figure 6—figure supplement 1F*). When mice were treated with PLX from P0 to P8 and allowed to age to P30, we found the cerebellar vermis section area was not decreased in PLX-treated IR mice compared to IR controls (*Figure 6N*). Thus, reducing the density of IBA1+ microglia/macrophages in neonatal mice reduces the recruitment of Nes-CFP+ cells to the EGL at P5 but does not have a long-term significant impact on regeneration of the cerebellum.

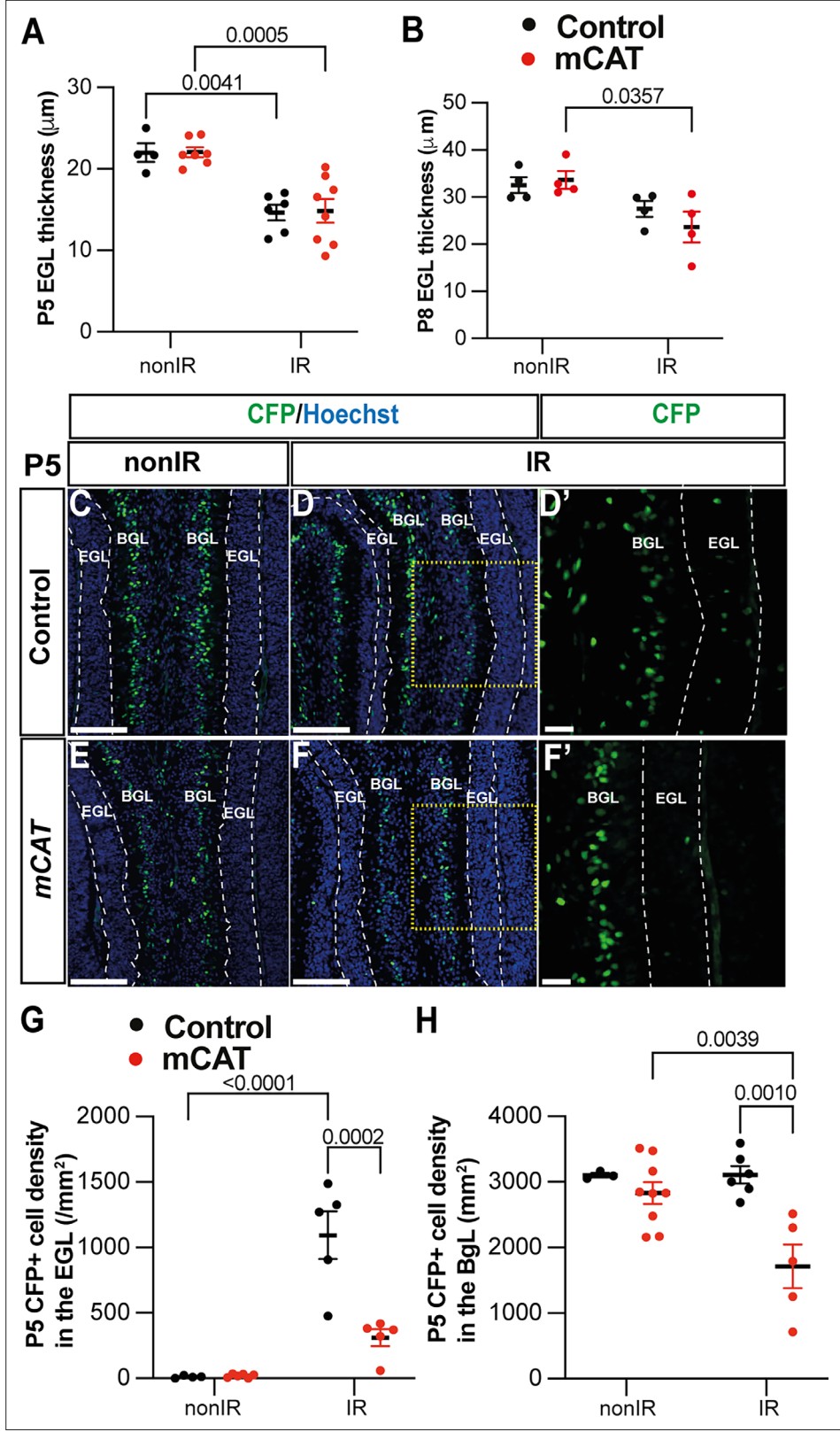

**Figure 5.** Reduced reactive oxygen species (ROS) impairs expansion of BgL-Nestin-expressing progenitors (NEPs) and their recruitment to the external granule layer (EGL) after injury. (**A, B**) Quantification of EGL thickness at P5 (two-way ANOVA, $F_{(1,21)}=36.64$, p<0.0001) (**A**) and P8 (two-way ANOVA, $F_{(1,12)}=11.34$, p=0.0056) (**B**) in lobules 3–5 of Nes-Cfp control and Nes-Cfp; mitochondrial catalase (mCAT) mutant mice with and without irradiation at P1. (**C–F**)

*Figure 5 continued on next page*

*Figure 5 continued*

Immunohistochemical (IHC) staining of medial sagittal cerebellar sections showing expression of CFP (green) in lobule 4/5 of Nes-Cfp control and Nes-Cfp; mCAT mutant mice at P5. Nuclei were counterstained with Hoechst (blue). (**D′**) and (**F′**) show high-power images of the yellow boxed area in the single channel CFP. EGL is delineated by the dashed white lines. (**G, H**) Quantification of CFP+ cell density in the EGL (two-way ANOVA, $F_{(1,19)}=5.192$, p=0.0359) (**G**) and BgL (two-way ANOVA, $F_{(1,17)}=6.191$, p=0.0223) (**H**) at P5 in Nes-Cfp control or Nes-Cfp; mCAT mutant non-irradiated and irradiated mice. EGL, external granular layer; BgL, Bergmann glia layer; P, postnatal day; nonIR, non-irradiated; IR, irradiated. Scale bar: D–F: 100 µm. Significant Tukey's post hoc multiple comparison tests are shown in the figures, and data are represented as mean ± SEM.

The online version of this article includes the following figure supplement(s) for figure 5:

**Figure supplement 1.** Reduced reactive oxygen species (ROS) impairs expansion of BgL-Nestin-expressing progenitors (NEPs) and their migration to the external granule layer (EGL) after injury.

## Discussion

We demonstrate that a transient increase in ROS signaling after cerebellar injury to the EGL is critical for adaptive reprogramming and full recovery of cerebellar growth. ROS likely acts as an alarm signal shortly after injury. scRNA-seq at P1–5 and bulk ATAC-seq at P2 of NEPs following targeted irradiation at P1 revealed a rapid increase in transcriptional and epigenetic changes associated with upregulation of ROS and stress-related pathways in NEPs that peaked at P2. A transient upregulation in ROS at P2 was confirmed using flow cytometry and found to correlate with the timing of cell death in the EGL 1 day after irradiation. By reducing mitochondrial ROS levels across all cell types at P2 using an mCAT transgene, we uncovered that ROS is required for several steps of adaptive reprogramming of BgL-NEPs. In addition, we found that microglia are reduced in injured mCAT pups, which is consistent with prior evidence that ROS can trigger immune cell recruitment in other systems (*Kim et al., 2010*; *Mehl et al., 2022*). Moreover, temporary depletion of microglia caused a reduction in the number of NEPs that migrate into the EGL at P5 following injury at P1, but no long-term reduction of cerebellar size. Thus, we identified key transcriptomic and epigenomic changes in cerebellar NEPs upon GCP ablation at birth and discovered roles for the tissue microenvironment, especially ROS and a more limited role of microglia during neonatal cerebellum regeneration.

scRNA-seq analysis of NEPs from nonIR (P1–3, P5) and nonIR (P2, P3, P5) showed an increase in genes associated with stress responses after irradiation in both the gliogenic and neurogenic subpopulations, but not in the GCPs. Furthermore, an increase in ROS signaling was detected 1 day after injury (P2) in all three lineages. The injury-induced ROS and stress-related gene signatures were not observed in the later P3+5 gliogenic- and neurogenic-NEPs, suggesting that the increase in ROS levels is an early injury-induced signal affecting the NEP transcriptome. Our bulk ATAC-seq data generated at P2 revealed that the open chromatin regions in the IR NEPs were enriched for transcription factor binding motifs related to ROS signaling and stress-induced transcription factors such as FOXO3 and AP1 (*Figure 1—source data 5*). Thus, cerebellar injury during development induces transcriptional and epigenomic signatures in the NEPs, and ROS signaling could be a key driver of NEP adaptive reprogramming. Our results are in line with other regeneration systems where a temporary increase in ROS is observed upon cell death or injury and is considered to be a DAMP (*Niethammer, 2016*).

Interestingly, although both gliogenic- and neurogenic-NEPs showed induction of cellular stress-related genes upon injury, upregulation of ROS signaling and related genes appeared greater in the *Hopx*-expressing gliogenic-NEPs that undergo adaptive reprogramming. Whether the upregulation of ROS signaling in the BgL-NEPs is due to BgL-NEPs being in proximity to the dying GCPs after injury or their direct contact due to their radial projections remains to be determined. The ability of BgL-NEPs to respond to GCP death via upregulating ROS signaling and impaired regeneration upon reduction of ROS levels shows that ROS signaling is involved in triggering adaptive reprogramming upon injury.

The cellular composition of the neonatal cerebellum is dramatically different from the adult. During the early postnatal period, we found that astrocytes in the WM below the lobules are the first to initiate GFAP expression at P2 and that by P5 all astrocytes express a high level of GFAP. In contrast, Bg express a low level of GFAP at P5 and reach a high level by P8. Interestingly, we found that injury leads to an increase in the level of GFAP expression in each type of astroglia compared to controls when they first initiate expression, deep WM astrocytes at P2 and Bg (and all astrocytes) at P5. Once

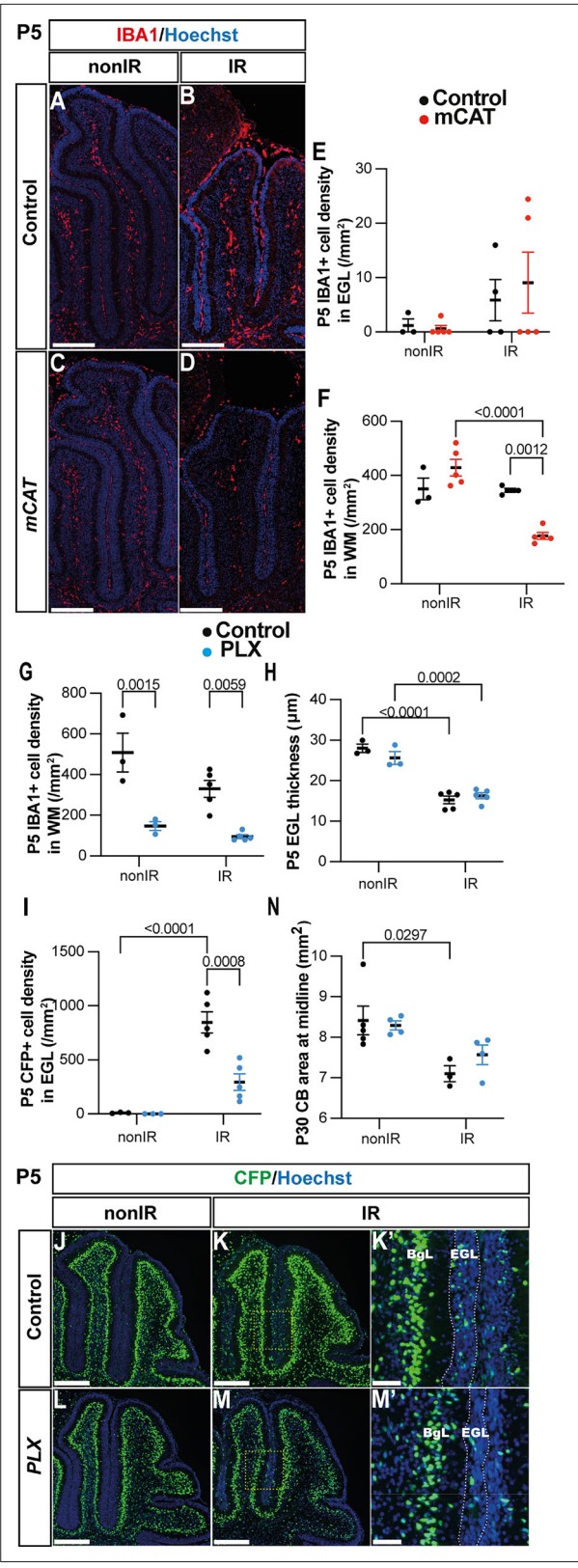

**Figure 6.** Microglia might promote recruitment of Nestin-expressing progenitors (NEPs) to the external granule layer (EGL) during cerebellar adaptive reprogramming after injury. (**A–D**) Immunohistochemical (IHC) staining of medial sagittal cerebellar sections for IBA1 (red) in control and mitochondrial catalase (mCAT) mice at P5. Nuclei were counterstained with Hoechst (blue). (**E, F**) Quantification of IBA1+ cell density in the external granular layer

*Figure 6 continued on next page*

*Figure 6 continued*

(**E**) and white matter (two-way ANOVA, $F_{(1,13)}$=24.74, p=0.0003) (**F**) at P5 on midsagittal sections of lobules 3–5 in the cerebellum of control and mCAT animals, with or without irradiation. (**G–J**) IHC staining of medial sagittal cerebellar sections at P5 for CFP (green) in lobule 4/5 of Nes-Cfp mice treated with PLX5622 or control DMSO with or without irradiation. Nuclei were counterstained with Hoechst (blue). (**H′**) and (**J′**) show a high-power image of the area indicated by yellow boxes. EGL is delineated by the white dashed lines. (**K**) Quantification of IBA1+ cell density in the white matter at P5 on midsagittal sections in lobules 3–5 of Nes-Cfp mice treated with PLX5622 or control DMSO, with or without irradiation (two-way ANOVA, $F_{(1,12)}$=42.40, p<0.001). (**L**) Quantification of EGL thickness at P5 in lobules 3–5 of Nes-Cfp mice treated with PLX5622 or control DMSO with or without irradiation (two-way ANOVA, $F_{(1,12)}$=109.5, p<0.001). (**M**) Quantification of CFP+ cells density in the EGL at P5 on midsagittal sections in lobules 3–5 of Nes-Cfp mice treated with PLX5622 or control DMSO with or without irradiation (two-way ANOVA, $F_{(1,12)}$=10.62, p=0.0068). (**N**) Measurement of cerebellar midsagittal section area at P30 in controls or mice treated with PLX, with or without irradiation at P1 (two-way ANOVA, $F_{(1,12)}$=13.29, p=0.0034). EGL, external granular layer; WM, white matter; P, postnatal day; nonIR, non-irradiated; IR, irradiated. Scale bar: A–D and G–J: 250 µm. Significant Tukey's post hoc multiple comparison tests are shown in the figures, and data are represented as mean ± SEM.

The online version of this article includes the following figure supplement(s) for figure 6:

**Figure supplement 1.** Microglia might promote recruitment of Nestin-expressing progenitors (NEPs) to the external granule layer (EGL) during cerebellar adaptive reprogramming after injury.

---

adaptive reprogramming is nearing completion (P8), the GFAP levels remained similar in all astroglia of nonIR and IR cerebella. These results suggest that the neonatal astrocytes respond to injury differently than in the adult where GFAP upregulation is observed immediately after injury (***Burda and Sofroniew, 2014***), since Bg and astrocytes in the lobules have a delayed response to injury with GFAP not being upregulated for several days. Furthermore, at birth, the microglia have not fully expanded in number in the cerebellum, and previous scRNA-seq showed that the neonatal and adult microglia are transcriptionally distinct (***Hammond et al., 2019***). Perhaps neonatal microglia are anti-inflammatory and pro-regenerative upon injury in neonates, in contrast to adult microglia, where upon traumatic brain injury, they can inhibit regeneration (***Donat et al., 2017***). Collectively, the differences in glial responses to injury in the neonatal cerebellum and adult brain likely contribute to the permissiveness of the neonatal cerebellum to regeneration. Details of the molecular changes that the neonatal cerebellar microglia/macrophages and astrocytes undergo upon GCP injury and their molecular crosstalk with NEPs remain to be determined.

We demonstrated the significance of ROS activation in the NEP reprogramming process using an mCAT transgenic mouse line in which human catalase (CAT) is expressed in mitochondria, a protein that can lead to a reduction in hydrogen peroxide ($H_2O_2$) and thus lower ROS. However, in mCAT neonatal cerebella, we found that the percentage of MitoSOX high cells was comparable to control mice under nonIR conditions. Importantly, however, 1 day after irradiation at P1 when the percentage of MitoSOX high cells increases in control IR mice, the mCAT transgene reduces the percentage of MitoSOX high cells, such that it remains at the baseline nonIR mCAT level. Therefore, human CAT expression in mitochondria in this model inhibits the injury-induced increase in ROS levels without affecting the homeostatic production of superoxide. Of possible relevance, in this mouse model, the observed change in ROS levels is likely global, impacting all cell types. The specific impact of increased mCAT in BgL-NEPs or microglia on their recruitment and function after an injury remains to be determined with new cell type-specific tools.

Our experiments depleting microglia using PLX5622 indicate that microglia/macrophages are involved in the regeneration of the EGL following irradiation. Previous studies have demonstrated a dual role of microglia in promoting and inhibiting regenerative processes within the nervous system (***Lee et al., 2021***; ***Wang et al., 2020***). Our data support the idea that neonatal microglia are involved in the adaptive reprogramming of NEPs to GCPs by promoting their replenishment of GCPs in the EGL. While the direct mechanisms remain to be discovered, given the small size of the lobules and disruption of the cytoarchitecture after injury, it might be possible for secreted factors from the WM microglia to reach the BgL NEPs. Alternatively, there could be a relay system through an intermediate cell type closer to the microglia. PLX treatment for 8 days after irradiation did not reduce the later growth of the injured cerebellum. One possibility is that after the cessation of PLX administration,

regeneration proceeds normally. Additionally, regenerative processes that act in parallel to microglial signaling are likely required.

Collectively, we have delineated the spatiotemporal cellular changes in the cerebellar glial microenvironment upon ablation of GCPs at birth and highlight ROS signaling as a key stimulator of adaptive reprogramming of NEPs. The details of how DAMP-glia-progenitor crosstalk is orchestrated remain to be untangled. Understanding how microenvironmental responses shape repair processes is a crucial first step toward developing strategies to promote regeneration.

## Acknowledgements

We thank past and present members of the Joyner laboratory for discussions and technical help. We would like to thank Dr. Ronan Chaligne and his team for their support in the multiplexed scRNA-seq experiments. We are grateful to the MSKCC Animal Imaging Core, Flow Cytometry Core, Center for Comprehensive Medicine and Pathology, Integrated Genomics Operation, Single-Cell Analytics and Innovation Laboratory, and Epigenetics Computational Laboratory teams for technical services and support. An XRad 225 Cx Microirradiator was purchased by support from a Shared Resources Grant from the MSKCC Geoffrey Beene Cancer Research Center. We gratefully acknowledge the support of the Gurdon Institute Scientific Computing Facility. This work was supported by grants from the NIH to ALJ (R01NS092096) and NSB (NINDS K99 NS112605-01). Additional funding was provided to ALJ from an NCI Cancer Center Support Grant (CCSG, P30 CA08748) and the Cycle for Survival, to SEN from a Francois Wallace Monahan Fellowship; to NSB from a Wellcome Career Development Award (227294/Z/23/Z), Royal Society grant (RGS\R1\231143) and Cambridge Stem Cell Institute Seed Funding, and to JBC from a University of Cambridge School of Biological Sciences DTP PhD Studentship and Peter and Emma Thomsen's Scholarship (1051). Gurdon Institute is supported by a Wellcome Core Grant (203144) and CRUK Grant (C6946/A24843).

---

## Additional information

### Funding

| Funder | Grant reference number | Author |
|---|---|---|
| National Institutes of Health | R01NS092096 | Alexandra L Joyner |
| National Institutes of Health | NS112605-01 | N Sumru Bayin |
| Wellcome Trust | 10.35802/227294 | N Sumru Bayin |
| National Institutes of Health | CA08748 | Alexandra L Joyner |
| Royal Society | RGS\R1\231143 | N Sumru Bayin |

The funders had no role in study design, data collection and interpretation, or the decision to submit the work for publication. For the purpose of Open Access, the authors have applied a CC BY public copyright license to any Author Accepted Manuscript version arising from this submission.

### Author contributions

Anna Pakula, Conceptualization, Formal analysis, Investigation; Salsabiel El Nagar, Formal analysis, Investigation, Writing – original draft, Writing – review and editing; N Sumru Bayin, Conceptualization, Formal analysis, Funding acquisition, Investigation, Writing – original draft, Writing – review and editing; Jens Bager Christensen, Formal analysis; Daniel Stephen, Formal analysis, Methodology; Adam James Reid, Data curation; Richard P Koche, Data curation, Formal analysis; Alexandra L Joyner, Conceptualization, Supervision, Funding acquisition, Writing – original draft, Project administration, Writing – review and editing

### Author ORCIDs

N Sumru Bayin ⓘ https://orcid.org/0000-0003-4371-855X

Richard P Koche https://orcid.org/0000-0002-6820-5083
Alexandra L Joyner https://orcid.org/0000-0001-7090-9605

### Ethics

All the mouse experiments were performed according to protocols approved by the Institutional Animal Care and Use Committee of Memorial Sloan Kettering Cancer Center (MSKCC) (protocol no. 07-01-001).

Reviewer #1 (Public review): https://doi.org/10.7554/eLife.102515.3.sa1
Reviewer #2 (Public review): https://doi.org/10.7554/eLife.102515.3.sa2
Author response https://doi.org/10.7554/eLife.102515.3.sa3

## Additional files

### Supplementary files

MDAR checklist

### Data availability

The scRNA-seq data was submitted to ArrayExpress (Accession E-MTAB-13353). Bulk ATAC-seq data has been submitted to GEO (GSE269342). The code used to carry out the scRNA-seq analysis can be found on GitHub (copy archived at *Christensen and Reid, 2025*).

The following datasets were generated:

| Author(s) | Year | Dataset title | Dataset URL | Database and Identifier |
|---|---|---|---|---|
| Pakula A, El Nagar S, Bayin NS, Christensen JB, Stephen DN, Reid AJ, Koche R, Joyner AL | 2024 | scRNA-seq time course of cerebellar Nestin-expressing progenitors from neonatal mice with and without injury at birth | https://www.omicsdi.org/dataset/biostudies-arrayexpress/E-MTAB-13353 | ArrayExpress, E-MTAB-13353 |
| Pakula A, El Nagar S, Bayin NS, Christensen JB, Stephen DN, Reid AJ, Koche R, Joyner AL | 2024 | An increase in ROS is required for repair of the neonatal cerebellum | https://www.ncbi.nlm.nih.gov/geo/query/acc.cgi?acc=GSE269342 | NCBI Gene Expression Omnibus, GSE269342 |

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

# Appendix 1

## Appendix 1—key resources table

| Reagent type (species) or resource | Designation | Source or reference | Identifiers | Additional information |
|---|---|---|---|---|
| Strain (*Mus musculus*) | *Nes-cfp* | Jackson Laboratories | Jax #034387 | |
| Strain (*Mus musculus*) | *Mcat* | Jackson Laboratories | Jax #016197 | |
| Chemical compound | 5-Ethynyl-2'-deoxyuridine (Edu) | Life Technologies | E10187 | |
| Chemical compound | Plx5622 | Plexxikon | Plexxikon via MTA | Currently commercially available |
| Antibody | Rabbit anti-Catalase | Athens Research and Technology | 01-05-030000 | 1/100 |
| Antibody | Chicken anti-GFAP | Abcam | ab4674 | 1/500 |
| Antibody | Rat anti-GFP | Nacalai Tesque | 440484 | 1/1000 |
| Antibody | Rabbit anti-IBA1 | Wako Chemicals | 019-19741 | 1/500 |
| Antibody | Goat anti-SOX2 | R&D Systems | Af2018 | 1/500 |
| Commercial assay kit | MitoSOX | Thermo Fisher Scientific | M36008 | |
| Commercial assay kit | MitoTracker | Thermo Fisher Scientific | M22426 | |
| Software, algorithm | Fiji (ImageJ) | NIH | | |
| Software, algorithm | FlowJo | BD Biosciences | | |
| Software, algorithm | Prism | GraphPad | | |

