## [Editor Report · eLife Assessment]

This **important** work substantially advances our understanding of reactive oxygen species (ROS) as a regenerative signal during postnatal cerebellum repair by activating adaptive progenitor reprogramming. The evidence supporting the conclusions is **compelling**, with rigorous genomic assays and in vivo analyses. This work will be of broad interest to biologists working on stem cells, neurodevelopment and regenerative medicine.

---

## [Referee Report · Reviewer #1 (Public review)]

Summary:

In this manuscript, Pakula et al. explore the impact of reactive oxygen species (ROS) on neonatal cerebellar regeneration, providing evidence that ROS activates regeneration through Nestin-expressing progenitors (NEPs). Using scRNA-seq analysis of FACS-isolated NEPs, the authors characterize injury-induced changes, including an enrichment in ROS metabolic processes within the cerebellar microenvironment. Biochemical analyses confirm a rapid increase in ROS levels following irradiation and forced catalase expression, which reduces ROS levels, and impairs external granule layer (EGL) replenishment post-injury.

Strengths:

Overall, the study robustly supports its main conclusion and provides valuable insights into ROS as a regenerative signal in the neonatal cerebellum.

Comments on revisions:

The authors have addressed most of the previous comments. However, they should clarify the following response:

*"For reasons we have not explored, the phenotype is most prominent in these lobules, that is why they were originally chosen. We edited the following sentence (lines 578-579):

First, we analyzed the replenishment of the EGL by BgL-NEPs in vermis lobules 3-5, since our previous work showed that these lobules have a prominent defect."*

It has been reported that the anterior part of the cerebellum may have a lower regenerative capacity compared to the posterior lobe. To avoid potential ambiguity, the authors should clarify that "the phenotype" and "prominent defect" refer to more severe EGL depletion at an earlier stage after IR rather than a poorer regenerative outcome. Additionally, they should provide a reference to support their statement or indicate if it is based on unpublished observations.

---

## [Referee Report · Reviewer #2 (Public review)]

Summary:

The authors have previously shown that the mouse neonatal cerebellum can regenerate damage to granule cell progenitors in the external granular layer, through reprogramming of gliogenic nestin-expressing progenitors (NEPs). The mechanisms of this reprogramming remain largely unknown. Here the authors used scRNAseq and ATACseq of purified neonatal NEPs from P1-P5 and showed that ROS signatures were transiently upregulated in gliogenic NEPs ve neurogenic NEPs 24 hours post injury (P2). To assess the role of ROS, mice transgenic for global catalase activity were assessed to reduce ROS. Inhibition of ROS significantly decreased gliogenic NEP reprogramming and diminished cerebellar growth post-injury. Further, inhibition of microglia across this same time period prevented one of the first steps of repair - the migration of NEPs into the external granule layer. This work is the first demonstration that the tissue microenvironment of the damaged neonatal cerebellum is a major regulator of neonatal cerebellar regeneration. Increased ROS is seen in other CNS damage models, including adults, thus there may be some shared mechanisms across age and regions, although interestingly neonatal cerebellar astrocytes do not upregulate GFAP as seen in adult CNS damage models. Another intriguing finding is that global inhibition of ROS did not alter normal cerebellar development.

Strengths:

This paper presents a beautiful example of using single cell data to generate biologically relevant, testable hypotheses of mechanisms driving important biological processes. The scRNAseq and ATACseq analyses are rigorously conducted and conclusive. Data is very clearly presented and easily interpreted supporting the hypothesis next tested by reduce ROS in irradiated brains.

Analysis of whole tissue and FAC sorted NEPS in transgenic mice where human catalase was globally expressed in mitochondria were rigorously controlled and conclusively show that ROS upregulation was indeed decreased post injury and very clearly the regenerative response was inhibited. The authors are to be commended on the very careful analyses which are very well presented and again, easy to follow with all appropriate data shown to support their conclusions.

Weaknesses:

The authors also present data to show that microglia are required for an early step of mobilizing gliogenic NEPs into the damaged EGL. While the data that PLX5622 administration from P0-P5 or even P0-P8 clearly shows that there is an immediate reduction of NEPs mobilized to the damaged EGL, there is no subsequent reduction of cerebellar growth such that by P30, the treated and untreated irradiated cerebella are equivalent in size. There is speculation in the discussion about why this might be the case. Additional experiments and tools are required to assess mechanisms. Regardless, the data still implicate microglia in the neonatal regenerative response, and this finding remains an important advance.

---

## [Author Response]

The following is the authors’ response to the current reviews.

**Reviewer #1 (Public review):**
Summary:In this manuscript, Pakula et al. explore the impact of reactive oxygen species (ROS) on neonatal cerebellar regeneration, providing evidence that ROS activates regeneration through Nestin-expressing progenitors (NEPs). Using scRNA-seq analysis of FACS-isolated NEPs, the authors characterize injury-induced changes, including an enrichment in ROS metabolic processes within the cerebellar microenvironment. Biochemical analyses confirm a rapid increase in ROS levels following irradiation and forced catalase expression, which reduces ROS levels, and impairs external granule layer (EGL) replenishment post-injury.Strengths:Overall, the study robustly supports its main conclusion and provides valuable insights into ROS as a regenerative signal in the neonatal cerebellum.Comments on revisions:The authors have addressed most of the previous comments. However, they should clarify the following response:*"For reasons we have not explored, the phenotype is most prominent in these lobules, that is why they were originally chosen. We edited the following sentence (lines 578-579):First, we analyzed the replenishment of the EGL by BgL-NEPs in vermis lobules 3-5, since our previous work showed that these lobules have a prominent defect."*It has been reported that the anterior part of the cerebellum may have a lower regenerative capacity compared to the posterior lobe. To avoid potential ambiguity, the authors should clarify that "the phenotype" and "prominent defect" refer to more severe EGL depletion at an earlier stage after IR rather than a poorer regenerative outcome. Additionally, they should provide a reference to support their statement or indicate if it is based on unpublished observations.

Our comment does not refer to a more severe EGL depletion at an earlier stage. There is instead poorer regeneration of the anterior region. The irradiation approach used provides consistent cell killing of GCPs across the cerebellum. This can be seen in Fig. 1c, e, g, i in our previous publication: Wojcinski, et al. (2017) Cerebellar granule cell replenishment post-injury by adaptive reprogramming of Nestin+ progenitors. Nature Neuroscience, 20:1361-1370. Also, Fig 2e, g, k, m in the paper shows that by P5 and P8, posterior lobule 8 recovers better than anterior lobules 1-5.

**Reviewer #2 (Public review):**
Summary:The authors have previously shown that the mouse neonatal cerebellum can regenerate damage to granule cell progenitors in the external granular layer, through reprogramming of gliogenic nestin-expressing progenitors (NEPs). The mechanisms of this reprogramming remain largely unknown. Here the authors used scRNAseq and ATACseq of purified neonatal NEPs from P1-P5 and showed that ROS signatures were transiently upregulated in gliogenic NEPs ve neurogenic NEPs 24 hours post injury (P2). To assess the role of ROS, mice transgenic for global catalase activity were assessed to reduce ROS. Inhibition of ROS significantly decreased gliogenic NEP reprogramming and diminished cerebellar growth post-injury. Further, inhibition of microglia across this same time period prevented one of the first steps of repair - the migration of NEPs into the external granule layer. This work is the first demonstration that the tissue microenvironment of the damaged neonatal cerebellum is a major regulator of neonatal cerebellar regeneration. Increased ROS is seen in other CNS damage models, including adults, thus there may be some shared mechanisms across age and regions, although interestingly neonatal cerebellar astrocytes do not upregulate GFAP as seen in adult CNS damage models. Another intriguing finding is that global inhibition of ROS did not alter normal cerebellar development.Strengths:This paper presents a beautiful example of using single cell data to generate biologically relevant, testable hypotheses of mechanisms driving important biological processes. The scRNAseq and ATACseq analyses are rigorously conducted and conclusive. Data is very clearly presented and easily interpreted supporting the hypothesis next tested by reduce ROS in irradiated brains.Analysis of whole tissue and FAC sorted NEPS in transgenic mice where human catalase was globally expressed in mitochondria were rigorously controlled and conclusively show that ROS upregulation was indeed decreased post injury and very clearly the regenerative response was inhibited. The authors are to be commended on the very careful analyses which are very well presented and again, easy to follow with all appropriate data shown to support their conclusions.Weaknesses:The authors also present data to show that microglia are required for an early step of mobilizing gliogenic NEPs into the damaged EGL. While the data that PLX5622 administration from P0-P5 or even P0-P8 clearly shows that there is an immediate reduction of NEPs mobilized to the damaged EGL, there is no subsequent reduction of cerebellar growth such that by P30, the treated and untreated irradiated cerebella are equivalent in size. There is speculation in the discussion about why this might be the case. Additional experiments and tools are required to assess mechanisms. Regardless, the data still implicate microglia in the neonatal regenerative response, and this finding remains an important advance.

As stated previously, the suggested follow up experiments while relevant are extensive and considered beyond the scope of the current paper.

The following is the authors’ response to the original reviews.

**Public Reviews:**

**Reviewer #1 (Public review):**
Summary:In this manuscript, Pakula et al. explore the impact of reactive oxygen species (ROS) on neonatal cerebellar regeneration, providing evidence that ROS activates regeneration through Nestin-expressing progenitors (NEPs). Using scRNA-seq analysis of FACS-isolated NEPs, the authors characterize injury-induced changes, including an enrichment in ROS metabolic processes within the cerebellar microenvironment. Biochemical analyses confirm a rapid increase in ROS levels following irradiation, and forced catalase expression, which reduces ROS levels, and impairs external granule layer (EGL) replenishment post-injury.Strengths:Overall, the study robustly supports its main conclusion and provides valuable insights into ROS as a regenerative signal in the neonatal cerebellum.Weaknesses:(1) The diversity of cell types recovered from scRNA-seq libraries of sorted Nes-CFP cells is unexpected, especially the inclusion of minor types such as microglia, meninges, and ependymal cells. The authors should validate whether Nes and CFP mRNAs are enriched in the sorted cells; if not, they should discuss the potential pitfalls in sampling bias or artifacts that may have affected the dataset, impacting interpretation.

In our previous work, we thoroughly assessed the transgene using RNA in situ hybridization for *Cfp*, immunofluorescent analysis for CFP and scRNA-seq analysis for *Cfp* transcripts (Bayin *et al*., *Science Adv*. 2021, Fig. S1-2)(1), and characterized the diversity within the NEP populations of the cerebellum. Our present scRNA-seq data also confirms that *Nes* transcripts are expressed in all the NEP subtypes. A feature plot for *Nes* expression has been added to the revised manuscript (Fig 1E), as well as a sentence explaining the results. Of note, since this data was generated from FACS-isolated CFP+ cells, the perdurance of the protein allows for the detection of immediate progeny of *Nes*-expressing cells, even in cells where *Nes* is not expressed once cells are differentiated. Finally, oligodendrocyte progenitors, perivascular cells, some rare microglia and ependymal cells have been demonstrated to express *Nes* in the central nervous system; therefore, detecting small groups of these cells is expected (2-4). We have added the following sentence (lines 391-394):

“Detection of *Nes* mRNA confirmed that the transgene reflects endogenous *Nes* expression in progenitors of many lineages, and also that the perdurance of CFP protein in immediate progeny of *Nes*-expressing cells allowed the isolation of these cells by FACS (Figure 1E)”.

(2) The authors should de-emphasize that ROS signaling and related gene upregulation exclusively in gliogenic NEPs. Genes such as Cdkn1a, Phlda3, Ass1, and Bax are identified as differentially expressed in neurogenic NEPs and granule cell progenitors (GCPs), with Ass1 absent in GCPs. According to Table S4, gene ontology (GO) terms related to ROS metabolic processes are also enriched in gliogenic NEPs, neurogenic NEPs, and GCPs.

As the reviewer requested, we have de-emphasized that ROS signaling is preferentially upregulated in gliogenic NEPs, since we agree with the reviewer that there is some evidence for similar transcriptional signatures in neurogenic NEPs and GCPs. We added the following (lines 429-531):

“Some of the DNA damage and apoptosis related genes that were upregulated in IR gliogenic-NEPs (*Cdkn1a*, *Phlda3*, *Bax*) were also upregulated in the IR neurogenic-NEPs and GCPs at P2 (Supplementary Figure 2B-E).”

And we edited the last few sentences of the section to state (lines 453-459):

“Interestingly, we did not observe significant enrichment for GO terms associated with cellular stress response in the GCPs that survived the irradiation compared to controls, despite significant enrichment for ROS signaling related GO-terms (Table S4). Collectively, these results indicate that injury induces significant and overlapping transcriptional changes in NEPs and GCPs. The gliogenic- and neurogenic-NEP subtypes transiently upregulate stress response genes upon GCP death, and an overall increase in ROS signaling is observed in the injured cerebella.”

(3) The authors need to justify the selection of only the anterior lobe for EGL replenishment and microglia quantification.

We thank the reviewers for asking for this clarification. Our previous publications on regeneration of the EGL by NEPs have all involved quantification of these lobules, thus we think it is important to stay with the same lobules. For reasons we have not explored, the phenotype is most prominent in these lobules, that is why they were originally chosen. We edited the following sentence (lines 578-579):

“First, we analyzed the replenishment of the EGL by BgL-NEPs in vermis lobules 3-5, since our previous work showed that these lobules have a prominent defect.”

(4) Figure 1K: The figure presents linkages between genes and GO terms as a network but does not depict a gene network. The terminology should be corrected accordingly.

We have corrected the terminology and added the following (lines 487-489):

“Finally, linkages between the genes in differentially open regions identified by ATAC-seq and the associated GO-terms revealed an active transcriptional network involved in regulating cell death and apoptosis (Figure 1K).”

(5) Figure 1H and S2: The x-axis appears to display raw p-values rather than log10(p.value) as indicated. The x-axis should ideally show -log10(p.adjust), beginning at zero. The current format may misleadingly suggest that the ROS GO term has the lowest p-values.

Apologies for the mistake. The data represents raw p-values and the x-axis has been corrected.

(6) Genes such as Ppara, Egln3, Foxo3, Jun, and Nos1ap were identified by bulk ATAC-seq based on proximity to peaks, not by scRNA-seq. Without additional expression data, caution is needed when presenting these genes as direct evidence of ROS involvement in NEPs.

We modified the text to discuss the discrepancies between the analyses. While some of this could be due to the lower detection limits in the scRNA-seq, it also highlights that chromatin accessibility is not a direct readout for expression levels and further analysis is needed. Nevertheless, both scRNA-seq and ATAC-seq have identified similar mechanisms, and our mutant analysis confirmed our hypothesis that an increase in ROS levels underlies repair, further increasing the confidence in our analyses. Further investigation is needed to understand the downstream mechanisms. We added the following sentence (lines 478-481):

“However, not all genes in the accessible areas were differentially expressed in the scRNA-seq data. While some of this could be due to the detection limits of scRNA-seq, further analysis is required to assess the mechanisms of how the differentially accessible chromatin affects transcription.”

(7) The authors should annotate cell identities for the different clusters in Table S2.

All cell types have been annotated in Table S2.

(8) Reiterative clustering analysis reveals distinct subpopulations among gliogenic and neurogenic NEPs. Could the authors clarify the identities of these subclusters? Can we distinguish the gliogenic NEPs in the Bergmann glia layer from those in the white matter?

Thank you for this clarification. As shown in our previous studies, we can not distinguish between the gliogenic NEPs in the Bergmann glia layer and the white matter based on scRNA-seq, but expression of the Bergmann glia marker *Gdf10* suggests that a large proportion of the cells in the *Hopx*+ clusters are in the Bergmann glia layer. The distinction within the major subpopulations that we characterized (*Hopx*-, *Ascl1*-expressing NEPs and GCPs) are driven by their proliferative/maturation status as we previously observed. We have included a detailed annotation of all the clusters in Table S2, as requested and a UMAP for *mKi57* expression in Fig 1E. We have clarified this in the following sentence (lines 383-385):

“These groups of cells were further subdivided into molecularly distinct clusters based on marker genes and their cell cycle profiles or developmental stages (Figure 1D, Table S2).”

(9) In the Methods section, the authors mention filtering out genes with fewer than 10 counts. They should specify if these genes were used as background for enrichment analysis. Background gene selection is critical, as it influences the functional enrichment of gene sets in the list.

As requested, the approach used has been added to the Methods section of the revised paper. Briefly, the background genes used by the *goseq* function are the same genes used for the probability weight function (*nullp*). The *mm8* genome annotation was used in the *nullp* function, and all annotated genes were used as background genes to compute GO term enrichment. The following was added (lines 307-308):

“The background genes used to compute the GO term enrichment includes all genes with gene symbol annotations within mm8.”

(10) Figure S1C: The authors could consider using bar plots to better illustrate cell composition differences across conditions and replicates.

As suggested, we have included bar plots in Fig. S1D-F.

(11) Figures 4-6: It remains unclear how the white matter microglia contribute to the recruitment of BgL-NEPs to the EGL, as the mCAT-mediated microglia loss data are all confined to the white matter.

We have thought about the question and had initially quantified the microglia in the white matter and the rest of the lobules (excluding the EGL) separately. However, there are very few microglia outside the white matter in each section, thus it is not possible to obtain reliable statistical data on such a small population. We therefore did not include the cells in the analysis. We have added this point in the main text (line 548).

“As a possible explanation for how white matter microglia could influence NEP behaviors, given the small size of the lobules and how the cytoarchitecture is disrupted after injury, we think it is possible that secreted factors from the white matter microglia could reach the BgL NEPs. Alternatively, there could be a relay system through an intermediate cell type closer to the microglia.” We have added these ideas to the Discussion of the revised paper (lines 735-738).

**Reviewer #2 (Public review):**
Summary:The authors have previously shown that the mouse neonatal cerebellum can regenerate damage to granule cell progenitors in the external granular layer, through reprogramming of gliogenic nestin-expressing progenitors (NEPs). The mechanisms of this reprogramming remain largely unknown. Here the authors used scRNAseq and ATACseq of purified neonatal NEPs from P1-P5 and showed that ROS signatures were transiently upregulated in gliogenic NEPs ve neurogenic NEPs 24 hours post injury (P2). To assess the role of ROS, mice transgenic for global catalase activity were assessed to reduce ROS. Inhibition of ROS significantly decreased gliogenic NEP reprogramming and diminished cerebellar growth post-injury. Further, inhibition of microglia across this same time period prevented one of the first steps of repair - the migration of NEPs into the external granule layer. This work is the first demonstration that the tissue microenvironment of the damaged neonatal cerebellum is a major regulator of neonatal cerebellar regeneration. Increased ROS is seen in other CNS damage models including adults, thus there may be some shared mechanisms across age and regions, although interestingly neonatal cerebellar astrocytes do not upregulate GFAP as seen in adult CNS damage models. Another intriguing finding is that global inhibition of ROS did not alter normal cerebellar development.Strengths:This paper presents a beautiful example of using single cell data to generate biologically relevant, testable hypotheses of mechanisms driving important biological processes. The scRNAseq and ATACseq analyses are rigorously conducted and conclusive. Data is very clearly presented and easily interpreted supporting the hypothesis next tested by reduce ROS in irradiated brains.Analysis of whole tissue and FAC sorted NEPS in transgenic mice where human catalase was globally expressed in mitochondria were rigorously controlled and conclusively show that ROS upregulation was indeed decreased post injury and very clearly the regenerative response was inhibited. The authors are to be commended on the very careful analyses which are very well presented and again, easy to follow with all appropriate data shown to support their conclusions.Weaknesses:The authors also present data to show that microglia are required for an early step of mobilizing gliogenic NEPs into the damaged EGL. While the data that PLX5622 administration from P0-P5 or even P0-P8 clearly shows that there is an immediate reduction of NEPs mobilized to the damaged EGL, there is no subsequent reduction of cerebellar growth such that by P30, the treated and untreated irradiated cerebella are equivalent in size. There is speculation in the discussion about why this might be the case, but there is no explanation for why further, longer treatment was not attempted nor was there any additional analyses of other regenerative steps in the treated animals. The data still implicate microglia in the neonatal regenerative response, but how remains uncertain.
**Recommendations for the authors:**

**Reviewer #2 (Recommendations for the authors):**
This is an exemplary manuscript.The methods and data are very well described and presented.I actually have very little to ask the authors except for an explanation of why PLX treatment was discontinued after P5 or P8 and what other steps of NEP reprogramming were assessed in these animals? Was NEP expansion still decreased at P8 even in the presence of PLX at this stage? Also - was there any analysis attempted combining mCAT and PLX?

We agree with the reviewer that a follow up study that goes into a deeper analysis of the role of microglia in GCP regeneration and any interaction with ROS signaling would interesting. However, it would require a set of tools that we do not currently have. We did not have enough PLX5622 to perform addition experiments or extend the length of treatment. Plexxikon informed us in 2021 that they were no longer manufacturing PLX5622 because they were focusing on new analogs for in vivo use, and thus we had to use what we had left over from a completed preclinical cancer study. We nevertheless think it is important to publish our preliminary results to spark further experiments by other groups.

References

(1) Bayin N. S. Mizrak D., Stephen N. D., Lao Z., Sims P. A., Joyner A. L. Injury induced ASCL1 expression orchestrates a transitory cell state required for repair of the neonatal cerebellum. Sci Adv. 2021;7(50):eabj1598.

(2) Cawsey T, Duflou J, Weickert CS, Gorrie CA. Nestin-Positive Ependymal Cells Are Increased in the Human Spinal Cord after Traumatic Central Nervous System Injury. J Neurotrauma. 2015;32(18):1393-402.

(3) Gallo V, Armstrong RC. Developmental and growth factor-induced regulation of nestin in oligodendrocyte lineage cells. The Journal of neuroscience : the official journal of the Society for Neuroscience. 1995;15(1 Pt 1):394-406.

(4) Huang Y, Xu Z, Xiong S, Sun F, Qin G, Hu G, et al. Repopulated microglia are solely derived from the proliferation of residual microglia after acute depletion. Nat Neurosci. 2018;21(4):530-40.